



# Exploring the elevated water vapor signal associated with the free-tropospheric biomass burning plume over the southeast Atlantic Ocean

Kristina Pistone[1,2], Paquita Zuidema[3], Robert Wood[4], Michael Diamond[4], Arlindo M. da Silva[5], Gonzalo Ferrada[6], Pablo Saide[7], Rei Ueyama[2], Ju-Mee Ryoo[8,2], Leonhard Pfister[2], James Podolske[2], David Noone[9,10], Ryan Bennett[1], Eric Stith[1,a], Gregory Carmichael[6], Jens Redemann[11], Connor Flynn[11], Samuel LeBlanc[1,2], Michal Segal-Rozenhaimer[1,2,12], and Yohei Shinozuka[1,2]

[1]Bay Area Environmental Research Institute, Moffett Field, CA, USA
[2]NASA Ames Research Center, Moffett Field, CA, USA
[3]University of Miami/Rosenstiel School of Marine and Atmospheric Science, Miami, FL, USA
[4]University of Washington, Seattle, WA, USA
[5]NASA Goddard Space Flight Center, Greenbelt, MD, USA
[6]Center for Global and Regional Environmental Research, The University of Iowa, Iowa City, IA, USA
[7]University of California, Los Angeles, Los Angeles, CA, USA
[8]Science and Technology Corporation, Moffett Field, CA, USA
[9]Department of Physics, University of Auckland, Auckland, NZ
[10]College of Earth, Ocean, and Atmospheric Sciences, Oregon State University, OR, USA
[11]University of Oklahoma, Norman, OK, USA
[12]Department of Geophysics, Porter School of the Environment and Earth Sciences, Tel-Aviv University, Tel-Aviv, Israel
[a]now at: JT4, LLC, Las Vegas, NV, USA

**Correspondence:** Kristina Pistone (kristina.pistone@nasa.gov)

**Abstract.** In southern Africa, widespread agricultural fires produce substantial biomass burning (BB) emissions over the region. The seasonal smoke plumes associated with these emissions are then advected westward over the persistent stratocumulus cloud deck in the Southeast Atlantic (SEA) Ocean, resulting in aerosol effects which vary with time and location. Much work has focused on the effects of these aerosol plumes, but previous studies have also described an elevated free-tropospheric water vapor signal over the SEA. Water vapor influences climate in its own right, and it is especially important to consider atmospheric water vapor when quantifying aerosol-cloud interactions and aerosol radiative effects. Here we present airborne observations made during the NASA ORACLES (ObseRvations of Aerosols above CLouds and their intEractionS) campaign over the SEA Ocean. In observations collected from multiple independent instruments on the NASA P-3 aircraft (from near-surface to 6-7km), we observe a strongly linear correlation between pollution indicators (carbon monoxide (CO) and aerosol loading) and atmospheric water vapor content, seen at all altitudes above the boundary layer. The focus of the current study is on the especially strong correlation observed during the ORACLES-2016 deployment (out of Walvis Bay, Namibia), but a similar relationship is also observed in the August 2017 and October 2018 ORACLES deployments.

Using ECMWF and MERRA-2 reanalyses and specialized WRF-Chem simulations, we trace the plume-vapor relationship to an initial humid, smoky continental source region, where it is subjected to conditions of strong westward advection, namely





the South African Easterly Jet (AEJ-S). Our analysis indicates that airmasses likely left the continent with the same relationship between water vapor and carbon monoxide as was observed by aircraft. This linear relationship developed over the continent due to daytime convection within a deep continental boundary layer (up to ∼5-6km) which produced fairly consistent vertical gradients in CO and water vapor, decreasing with altitude and varying in time, but does not originate as a product of the BB

combustion itself. Due to a combination of conditions and mixing between the smoky, moist continental boundary layer and the dry and fairly clean upper-troposphere air above (∼ 6km), the smoky, humid air is transported by strong zonal winds and then advected over the SEA (to the ORACLES flight region) following largely isentropic trajectories. HYSPLIT back trajectories support this interpretation. Better understanding of the conditions and processes which cause the water vapor to covary with plume strength is important to accurately quantify direct, semi-direct, and indirect aerosol effects in this region.

**1   Introduction**

Biomass burning (BB) is a substantial global source of absorbing aerosols, and the effect of these aerosols is a subject of much study in climate science. The cumulative climatic impact of aerosols is a significant source of uncertainty in our present understanding of the earth system (Boucher et al., 2013), and the question is further complicated when one considers absorbing aerosols, which, rather than solely scattering sunlight, can also absorb solar radiation, causing a local heating effect (Myhre

et al., 2013). The manifestation of the so-called semi-direct aerosol effects are known to be linked to both meteorological regime and to the relative location of aerosol and cloud within the atmosphere (Koch and Del Genio, 2010). Thus absorbing aerosols, such as those produced by biomass burning, may influence atmospheric dynamics and cloud properties through local heating/cloud burnoff and/or by reducing or enhancing atmospheric convection, but the aerosol effects may also be driven by these same radiative or meteorological factors. Biomass burning not only emits aerosols but also produces gaseous components

such as carbon monoxide (CO), which can be used as an indicator of air mass origin as it is not affected by aerosol aging or removal processes.

Previous studies have documented higher amounts of water vapor over the southeast Atlantic (SEA) during the BB season. Adebiyi et al. (2015) co-located MODIS satellite retrievals of AOD with radiosondes out of St Helena Island (15.9°S, 5.6°W), and found that free-tropospheric aerosol transported from the African continent was associated with elevated moisture content

between 750 and 500 hPa (∼2.5-6km). This is important to understand as humidity, particularly if it's co-located with absorbing aerosols, will affect the radiative profile of the atmosphere and underlying cloud properties. Indeed, the authors concluded that the elevated moisture observed over St Helena increased shortwave heating to a small degree and had a larger impact of increasing longwave cooling: the maximum net LW cooling due to water vapor near the top of the layer reduced the impact of shortwave aerosol absorption by approximately a third (Adebiyi et al., 2015). In an earlier, more general study, Ackerman

et al. (2004) worked to quantify the effects of water vapor by modeling the influence of above-cloud water vapor using several case studies informed by field measurements. The authors concluded that the cloud liquid water path response to aerosol (via aerosol indirect effects) had a much stronger response in the presence of overlying water vapor than under dry conditions. Later, Wilcox (2010) used satellite observations over the SEA to determine that the presence of aerosol above cloud increased





cloud liquid water path, which the author attributed to a radiative stabilization of the boundary layer. Adebiyi and Zuidema (2018) also showed that moisture at 600 hPa was negatively correlated with low cloud cover, attributed to a reduction of cloud top cooling, although a recent paper by Scott et al. (2020) found a positive correlation between moisture at 700 hPa and low cloud cover, which they attribute to the entrainment of moisture helping to support the cloud deck. Moisture changes at 700

and 600 hPa can be anti-correlated in this region, allowing a reconciliation of these results.

Even aside from cloud-related effects, elevated water vapor will have impacts on radiative transfer through the atmospheric column, both in terms of shortwave heating and longwave cooling (as was described by Adebiyi et al., 2015). The authors of that study also showed that differences in longwave cooling were more strongly associated with the free-tropospheric water vapor signal than with the cloud thickness itself, which illustrates the strong radiative potential of water vapor in this region.

Recently, Marquardt Collow et al. (2020) used data from the LASIC (Layered Atlantic Smoke Interactions with Clouds) field campaign based at Ascension Island (7.96°S, 14.35°W) in conjunction with MERRA-2 reanalysis and a radiative transfer model to quantify the radiative heating rate due to aerosols and clouds for July through October of 2016 and 2017. They found strong cloud-top longwave cooling and strong cloud-top shortwave heating due to absorbing aerosols, with a monthly mean maximum heating rate of 2.1-2.4K/d in September 2016 (approximately double the heating rate found by Adebiyi et al. (2015)

over St Helena Island). In this study the authors noted that an increase in relative humidity around 700hPa was coincident with the appearance of the aerosol plume and accounted for this in determination of the aerosol optical properties, but did not explicitly consider the impacts of the co-located humidity in these profiles.

Finally, Deaconu et al. (2019) used CALIPSO, POLDER, and MODIS satellite data in conjunction with ERA-Interim re-analysis fields over the SEA, and found that the presence of water vapor reduces longwave cloud top cooling, potentially

causing thicker clouds to develop. We note that this work focused on June-July-August, which has substantial meteorological differences versus September-October; specifically, the moisture levels at 700hPa are lower than 2.5g/kg in June and July, in contrast with values around 5g/kg in August and September (Deaconu et al., 2019). A new study by Baró Pérez et al. (2020) also used satellite observations and reanalysis to study the impact of aerosol type on heating in the SEA and found water vapor to be associated with aerosol layers, but interestingly found a significant and negative correlation between AOD and relative

humidity during September and October. Nonetheless, these works establish the importance of the humid layer to the radiative balance of the aerosol-cloud system in the SEA, and Deaconu et al. (2019) in particular suggests that the impacts in the later, more humid months could be enhanced relative to what has been calculated for JJA.

Taken together, these previous studies suggest that, first, the presence of above-cloud water vapor in conjunction with aerosol may modify the underlying cloud properties without physically mixing into the cloud layer to alter the microphysics; and,

second, the presence of water vapor associated with the presence of absorbing aerosol will have impacts on both longwave and shortwave radiation in the atmospheric column. The amount of time this above-cloud vapor is co-located with above-cloud smoke will determine the ultimate magnitude of these effects over the SEA as a whole. Thus, it is of interest to explore the sources and airmass history of this smoky, humid layer over the SEA.



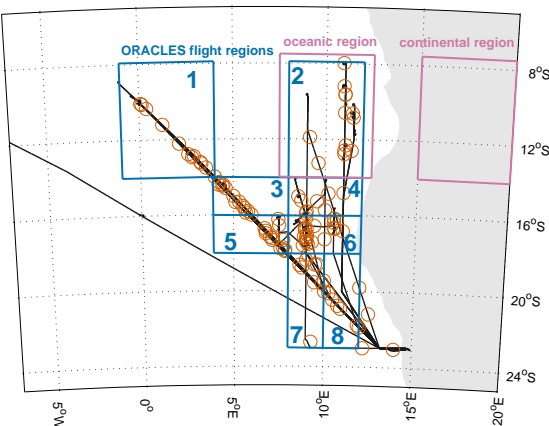

**Figure 1.** Map showing the flight tracks of the 14 P-3 flights during ORACLES-2016 (black lines) and the areas of study in this work. Note that the SE-to-NW diagonal (passing through Zones 1, 3, 5, 7, and 8) includes six "routine flights" overlaying one another. Reddish circles indicate locations of the 95 partial or full aircraft vertical profiles during all flights which will be discussed in more detail in Section 3.3. The blue boxes indicate the regional subsets (labeled Zones 1-8) used in the spatially-subdivided aircraft analysis in Section 3, and the lavender boxes show the oceanic and continental regions used for the reanalysis analysis in Section 3.4.

In this paper, we use recent aircraft measurements over the SEA Ocean, combined with large-scale meteorological reanalyses and specialized models, to identify and explore this feature of co-located humidity and BB plume. With the new aircraft-based observations discussed here, we are able to gain a better understanding of this relationship than was previously possible.

In the bulk of our analysis, we use carbon monoxide (CO) as a tracer of biomass burning emissions. CO is not aged or removed by cloud processes as the BB aerosols are, and thus is a more reliable indicator to determine air mass origin than the aerosols themselves. Modeled CO is also more robust than modeled outputs of individual aerosol species (e.g., Shinozuka et al., 2020), and thus allows analysis of airmass origins and trajectories by comparing to these products. However, the results we show using observed CO are largely consistent with results using aircraft-measured aerosol extinction or scattering.

The NASA ORACLES (ObseRvations of Aerosols above CLouds and their intEractionS) campaign was a 5-year, multi-institutional project to study the effects of biomass burning aerosols and their interactions with the southeast Atlantic stratocu-mulus deck (Zuidema et al., 2016; Redemann et al., 2020). ORACLES had three field deployments during the African biomass burning season: out of Walvis Bay, Namibia, in September 2016, and out of São Tomé, São Tomé e Príncipe in August 2017 and October 2018. Each of these deployments used a NASA P-3 aircraft for tropospheric sampling (roughly 0-7km), although



the 2016 deployment had an additional high-flying ER-2 aircraft (above 20km) for downward-looking remote sensing measurements. There are significant logistical and meteorological differences between each deployment; due to the different seasonal timing (by design of the ORACLES campaign) and the different deployment locations of 2016 versus 2017/2018, we analyze each deployment separately. In this work, we focus on data from the P-3 aircraft during the September 2016 deployment, with

some discussion of the August 2017 and October 2018 observations to provide insight into the multi-year context. A more detailed discussion of the three ORACLES deployments may be found in Redemann et al. (2020). Figure 1 shows all P-3 flight paths and the locations of aircraft profiles for 2016 (i.e., the main focus of the present paper), as well as some key spatial delineations which we will use.

In Section 2, we introduce the instruments, data, and reanalysis and model products used. In Section 3.1, we present analysis

of the atmospheric humidity as measured by three independent instruments aboard the P-3 aircraft, and in Section 3.2 we discuss how the water vapor relates to the presence of the biomass burning plume over the SEA in ORACLES-2016. We next compare our observations to reanalysis products and model outputs over the SEA (Section 3.3), and over the continental source region (Section 3.4). In Section 3.5 we briefly discuss the 2017 and 2018 results and how they differ from the 2016 deployment. In Section 4, we synthesize the results before discussing potential causes of the observed patterns, their context within previous

studies of the region, and their potential radiative implications.

## 2    Instruments and Methods

In this work we use observational data from ORACLES in conjunction with large-scale atmospheric reanalysis and the outputs of specialized model configurations, as described below.

### 2.1    Aircraft instrumentation

The observational data considered here are from the ORACLES dataset. The full dataset is archived at https://doi.org/10.5067/ Suborbital/ORACLES/P3/2016_V1 for the 2016 deployment, https://doi.org/10.5067/Suborbital/ORACLES/P3/2017_V1 for 2017 and https://doi.org/10.5067/Suborbital/ORACLES/P3/2018_V1 for 2018. All instruments used here were deployed on the P-3 aircraft during all three ORACLES deployments. 1 Hz measurements are used unless otherwise indicated. Individual flights were classified as either "routine flights," which in 2016 extended along a diagonal flight path from (20°S, 10°E) to (10°S,

0°E), or "flights of opportunity," which focused on specific science objectives and were largely nearer to the Namibian/Angolan coast (Figure 1). A more complete overview of the ORACLES operations and major results can be found in Redemann et al. (2020).

### 2.1.1    4STAR

The Spectrometer for Sky-Scanning Sun-Tracking Atmospheric Research (4STAR; Dunagan et al., 2013) is an airborne hy-

perspectral (350-1700 nm) sun photometer which can make direct-beam (sun-tracking mode) measurements for retrieval of column aerosol optical depth (AOD; e.g., Shinozuka et al., 2013) and column trace gases (e.g., Segal-Rosenheimer et al.,





2014) above the aircraft level. This work presents the column AOD and column water vapor (CWV) measured by 4STAR; additional ORACLES-2016 results using 4STAR measurements may be found in LeBlanc et al. (2020) for AOD, and Pistone et al. (2019) for airborne retrievals of aerosol intensive properties using AERONET-like radiance inversions.

### 2.1.2 COMA

In all ORACLES deployments, volume mixing ratios of carbon monoxide (CO), carbon dioxide ($CO_2$), and water vapor ($q$) were measured by a Los Gatos Research $CO/CO_2/H_2O$ Analyzer (known as COMA), modified for flight operations. It uses off-axis integrated cavity output spectroscopy (ICOS) technology to make stable cavity enhanced absorption measurements of CO, $CO_2$, and $H_2O$ in the infrared spectral region, technology that previously flew on other airborne research platforms with a precision of 0.5 ppbv over 10s (Liu et al., 2017; Provencal et al., 2005). Water vapor measurements of less than 50 ppmv
($\sim$0.03 g/kg) were removed due to instrument limitations, but this has minimal effect on the data considered here.

The CO measured during ORACLES is used in the present work as a tracer for air masses originating from combustion. While a major focus of ORACLES is the radiative effects of aerosols, CO will be conserved even under cloud processing which may affect the aerosol concentrations from biomass burning, and thus provides valuable information on air mass origin (and simplifies the comparison to modeled parameters).

### 2.1.3 WISPER

Atmospheric water vapor was also measured as part of the Water Isotope System for Precipitation and Entrainment Research (WISPER) data (reporting $H_2O$ concentration, D/H and $^{18}O/^{16}O$ isotope ratios). For ORACLES WISPER was continued to use a pair of gas phase isotopic analyzers based on the Picarro Incorporated L2120-i Water Vapor Isotopic Analyzer (Gupta et al., 2009). Coupled to the near-isokinetic SDI inlet, the system reports total water (vapor plus condensate), which can be
interpreted as vapor when out of cloud. Air was sampled from the inlet flow at 2.5 slpm via a 6 meter long thermally-insulated copper transfer line heated to 50°C to minimize any wall effects and avoid possible condensation in the lines. The exterior portion of the SDI inlet was unheated. Two different Picarro L2120-i instruments were used during the 2016 campaign, one for the dates up to and including 04 September, and another for later dates. The switch was associated with an instrument failure that led to poor data recovery on 3 of the 14 flights (Table 1). The instrument used in the first part of the campaign reports
data at 5Hz while the instrument used later in the campaign reports at 0.5 Hz. Both data are aggregated onto a 1Hz common time using simple binning, and synchronized to the data system using cloud probes timing when entering/exiting clouds. Time synchronization has an uncertainty of about 1 second. Calibration of the system based on pre-campaign lab calibration using a LI-COR Model 610 dew point generator at a fixed temperature, with air diluted with ultra-zero grade dry air to span low concentration range using quantitatively calibrated mass flow controllers. The water vapor measurements are valid to 10ppmv
(0.016 g/kg) and precision was typically reported as between 9-50ppmv (0.01-0.08 g/kg), with greater values corresponding to lower absolute water vapor amount.





### 2.1.4 P-3 aircraft data

The P-3 aircraft is equipped with instrumentation to make a number of standard on-board measurements of environmental data such as temperature, pressure, relative humidity, and wind speed. A full description of the on-board instrumentation may be found in Section 4.6 of the aircraft handbook at https://airbornescience.nasa.gov/sites/default/files/P-3B%20Experimenter%20Handbook%20548-HDBK-0001.pdf. The aircraft-based specific humidity ($q$) considered here was calculated from the reported dew point temperature (from an EdgeTech Model 137 aircraft dew point hygrometer) and static pressure (from a Rosemount MADT 2014 sensor) values following Vaisala (2013):

$$q = \frac{p_{ws}}{(p_{meas} - p_{ws})} \times m_r \times 10^3 \text{ (in g/kg)},\tag{1}$$

where $p_{meas}$ is the measured static pressure, $m_r$ is the ratio of the molecular weight of water vapor to air (18.015/28.97), and $p_{ws}$ is the simplified formula for water vapor saturation pressure over water given as

$$p_{ws} = A \times 10^{mT_{dp}/(T_{dp}+T_n)},\tag{2}$$

where $T_{dp}$ is the measured dew point temperature and the constants $A$, $m$, and $T_n$ are $6.116441$ hPa, $7.591386$, and $240.7263°$C, respectively (Vaisala, 2013). The static pressure measurements have a precision of 0.5 hPa and an accuracy of $\pm 2.5$ hPa. For the dew point hygrometer, measurement precision was $0.1°$C and an accuracy of $0.2°$C nominally, with greater uncertainty below $0°$C and during profiles with large $\delta T_{dp}/\delta T$.

## 2.2 Large-scale reanalyses and models

In conjunction with these observations, we select two large-scale reanalyses, which assimilate satellite observations and thus should be consistent with conditions observed by aircraft; and two free-running models, which, due to their unconstrained nature, may help to diagnose which processes are/not in play. The reanalyses considered are the latest iteration of the European Centre for Medium-Range Weather Forecasts (ECMWF) reanalysis, ERA5 (CDS, 2017) and NASA's Modern-Era Retrospective analysis for Research and Applications, Version 2 (MERRA-2; Gelaro et al., 2017). The former was chosen due to its exceptionally good agreement with the ORACLES observations (Section 3.3), and the latter was chosen as it incorporates aerosol observations, the lack of which is a shortcoming of the ERA product. We also briefly show results using the previous ERA-Interim (Dee et al., 2011) for continuity with previous work.

For the models, we consider two different specialized configurations of WRF developed in support of the ORACLES mission, termed WRF-CAM5 and WRF-Chem for consistency with previous studies (e.g., Shinozuka et al., 2020). The similarities and differences between each of these products is not the focus of the present paper, but the results of the differences between each product allows us to diagnose the influence of potential drivers in the real world.

### 2.2.1 ECMWF reanalyses

The European Centre for Medium-Range Weather Forecasts (ECMWF) has developed global atmospheric reanalysis products for several decades, with the ERA-Interim (Dee et al., 2011) serving as the primary reanalysis product through mid-2019, before



being surpassed by the recently-released ERA5 (Hersbach et al., 2019). ERA5 is considered at 0.25-degree, hourly resolution in the comparison with ORACLES flights (Section 3.3), and 0.25-degree, 3-hourly resolution in the continental analysis (Sections 3.4 and 4.1). ERA5 does not report atmospheric chemistry or aerosols nor does it directly incorporate aerosol effects, though satellite measurements of aerosol-influenced radiances are incorporated into the reanalysis. ERA-Interim was only available at

3-hourly resolution. Due to the timing in the ERA5 dataset release, we explored results using both of these products in Section 3.3, and found ERA5 performs generally better compared to the observations. In Supplementary Materials (Figures S1 and S2) we provide selected comparisons between ERA5, ERA-Interim, and observations over the SEA.

### 2.2.2 MERRA-2

The Modern-Era Retrospective Analysis for Research and Applications, version 2 (MERRA-2) is an atmospheric reanalysis

produced by NASA's Global Modeling and Assimilation Office (GMAO) (Gelaro et al., 2017; Randles et al., 2017; Buchard et al., 2017). MERRA-2 assimilates observations of meteorological parameters from multiple satellite platforms, as well as aerosol optical depth from satellites (MODIS, AVHRR) and ground-based (AERONET) measurements, into a comprehensive atmospheric model, with explicit accounting of aerosol radiative effects. MERRA-2 datasets are given on a nominal 50 km horizontal resolution ($0.5° \times 0.5°$) with 72 vertical layers from the surface to 0.01 hPa. An additional goal of the ORACLES

campaign was to evaluate chemical transport models and reanalysis products such as MERRA-2, and to this end the complete set of MERRA-2 files have been sampled up to 1-second resolution along every ORACLES flight (Collow et al., 2020). These products are available online at https://portal.nccs.nasa.gov/datashare/iesa/campaigns/ORACLES/. Over the larger continental and oceanic domain, both MERRA-2 and ERA5 are considered at 3-hourly temporal resolution.

### 2.2.3 WRF-CAM5

The WRF-CAM5 configuration was run at 36km horizontal resolution over the month of September 2016, with 72 vertical layers (50 layers below 3km) with a domain of 14°N to 41°S and 34°W to 51°E. It used CAM5 aerosol and physics, with MAM3 aerosols and CESM cloud microphysics and cumulus, with shallow cumulus turned off. Smoke emissions were from QFED, with no inversion and no plume rise. This model was initialized every 5 days, with 2 days spin-up for each initialization (i.e., 3-day continuous runs at a time). Aerosol initial conditions were from the previous cycle, while the meteorology for

each initialization was from NCEP-FNL-ANL, with chemistry and aerosols from CAMS reanalyses. We also note that here the ORACLES along-track WRF-CAM5 outputs are used at 10s resolution. A more detailed description can be found in Shinozuka et al. (2020).

### 2.2.4 WRF-Chem

The WRF-Chem simulations were performed for the period of 15 August to 30 September 2016 at 28 km resolution and 67

vertical levels covering a domain from 13.9°N to 35.6°S and 26.5°W to 42.5°E. Daily QFED biomass burning emissions were used following a diurnal cycle with a maximum at 2 PM local time (normal distribution), with additional EDGAR HTAP




(Emissions Database for Global Atmospheric Research Hemispheric Transport of Air Pollution) anthropogenic and MEGAN (Model of Emissions of Gases and Aerosols from Nature) biogenic emissions. Radiation and aerosol-meteorology feedback were turned on, and a smoke plume rise process was enabled.

Initial and boundary conditions from ERA5 and CAMS reanalysis were used to account for the meteorology and chemistry and aerosols, respectively. Simulations were initialized every day at 00Z and ran for 30 hours. The first 6 hours were discarded to account for the meteorology spin up. We consider the period between 15-31 August (17 days) as a spin up for chemistry and aerosols. CAMS was used for boundary conditions during the whole simulation to account for possible intrusion of aerosols outside the domain (e.g. Saharan dust, smoke from Madagascar, sea salt). CAMS was used only for the 15 August initialization, and subsequent simulations were initialized by recycling the chemistry and aerosols from the previous run. In this manner, we can assume that all chemistry and aerosols used here are explicitly calculated by the model. In contrast, ERA5 was used for initialization and boundary conditions throughout the whole simulation (i.e., at daily reinitialization).

### 2.2.5 NOAA HYSPLIT trajectories

We ran NOAA's Hybrid Single-Particle Lagrangian Integrated Trajectory model (HYSPLIT; Stein et al., 2016) to trace air-masses sampled by aircraft profiles towards their origins. Runs were computed offline using a standard HYSPLIT back-trajectory configuration. As ERA5 is not currently available as a HYSPLIT meteorological input, the meteorology used is from the National Centers for Environmental Prediction (NCEP) Global Data Assimilation System (GDAS) 0.5-degree model, provided directly by NOAA HYSPLIT (ftp://arlftp.arlhq.noaa.gov/pub/archives/gdas0p5), which is the highest resolution available for 2016. Trajectories are run using vertical motion determined alternately by the default "model motion" (kinematic trajectories using winds from the GDAS meteorology) and using isentropic pathways calculated from GDAS potential temperature fields (https://www.arl.noaa.gov/documents/workshop/NAQC2007/HTML_Docs/trajvert.html).

## 3 Results

### 3.1 Measured humidity from different ORACLES instruments

Before presenting the analysis of the BB plume as it relates to the humidity, we first show the robustness of the water vapor measurements by comparing the three independent instruments available during ORACLES: COMA, WISPER, and the dewpoint hygrometer ("Aircraft") data from an onboard hygrometer (Table 1), as were described in Section 2.

Figure 2 shows measurements from the three water vapor instruments for all 2016 flights at 1s resolution, for the full dataset and for specific subsets based on altitude (i.e., excluding layers which are clearly boundary layer altitudes) or water vapor gradient (i.e., to minimize the effect of varying instrument response times). The correlations in all cases are robust and statistically significant ($R^2 > 0.97$ for all data; Table 2), likely due in part to the large amount of data collected. Significant deviations from the 1:1 line (grey dots) occur either during high humidity conditions within the planetary boundary layer, or during a rapid change in water vapor conditions, which can be explained in part as due to inlet differences and related issues





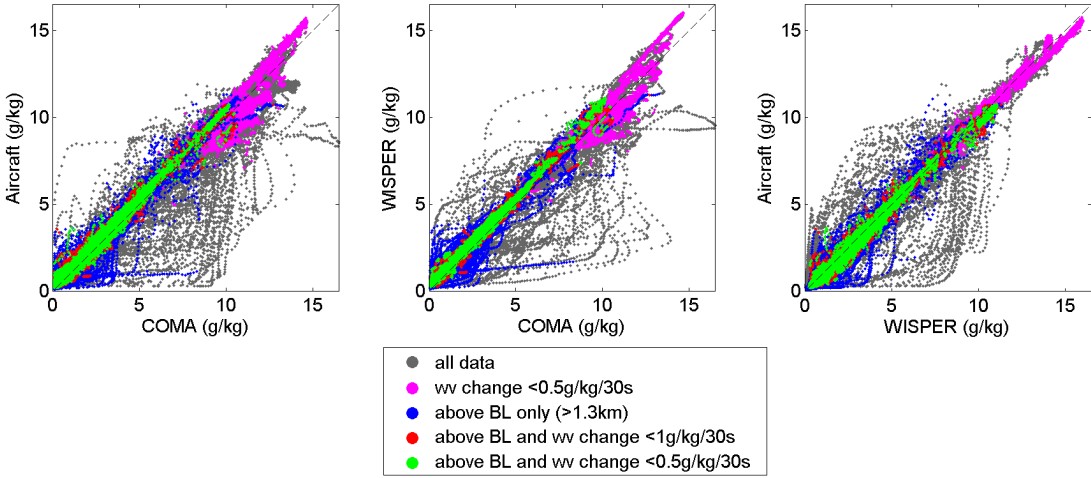

**Figure 2.** Comparison of water vapor specific humidity $q$ from the three instruments for ORACLES-2016, for all flight data and for subsets based on altitude or change of water vapor with altitude. This subsetting highlights that the majority of the disagreement between instruments (grey dots) is either within the planetary boundary layer and/or during aircraft ascents/descents through rapidly changing conditions.

**Table 1.** Data availability from each in-situ instrument, as a percentage of all flight time (takeoff-landing), 30 August 2016-27 September 2016, and as a fraction of only in-plume level leg or vertical profile flight time (40% of total flight time). A large portion of instrument downtime was during transit periods above the plume level.

| Instrument | Flights available | % uptime (total) | % uptime (plume/profiles) |
|---|---|---|---|
| COMA CO | all | 99.5% | 99.98% |
| COMA water vapor | all | 59.1% | 87.0% |
| Aircraft water vapor | all | 98.5% | 99.1% |
| WISPER water vapor | 11/14 | 70.9% | 76.4% |

of differing instrument response times. Deviations are expected during transitions from in-cloud to out-of-cloud conditions since each inlet system has differing heating to manage (or otherwise avoid) condensation artifacts. Focusing specifically on the in-plume conditions, we find that the instruments shows quantitatively consistent water vapor measurements, with slopes of total least-squares fits between 0.98 and 1.01. These strong correlations between independent instruments on the same platform

5   indicate that the observed water vapor signal is robust.

Having established that we have good confidence in the robustness of the water vapor data measured by multiple instruments, in the following sections, we focus largely on COMA water vapor. This instrument measures $q$ with greater precision than the aircraft probe, and more data are available from COMA than from WISPER for flight times either within the biomass burning plume, or profiling the full atmosphere (Table 1), which are the sampling times on which we focus in this study. Additionally,

10  while temporal corrections have been applied to synchronize the various instruments against one another, COMA CO and $q$



**Table 2.** Correlations between measures of water vapor for ORACLES 2016, 1s data resolution, for the subsets shown in Figure 2. All correlations are significant at the $p < 0.001$ level; the statistical significance and the slopes of the correlations are largely similar for each subset.

| Subset | COMA vs Aircraft | | | COMA vs WISPER | | | WISPER vs Aircraft | | |
|---|---|---|---|---|---|---|---|---|---|
| | fit | $R^2$ | # pts | fit | $R^2$ | # pts | fit | $R^2$ | # pts |
| All data | 0.98x+0.21 | 0.973 | 271932 | 0.98x+0.33 | 0.977 | 217675 | 0.993x-0.075 | 0.988 | 288992 |
| z> 1.3km | 1.01x+0.15 | 0.984 | 174756 | 1.01x+0.29 | 0.987 | 143437 | 0.985x-0.068 | 0.990 | 210101 |
| z> 1.3km; $\Delta q$ <1.0g/kg/(30s) | 1.01x+0.17 | 0.987 | 130181 | 1.01x+0.28 | 0.995 | 130207 | 0.995x-0.108 | 0.990 | 130720 |
| z> 1.3km; $\Delta q$ <0.5g/kg/(30s) | 1.01x+0.18 | 0.991 | 118227 | 1.01x+0.29 | 0.996 | 118231 | 0.995x-0.108 | 0.992 | 118720 |
| All z; $\Delta q$ <0.5g/kg/(30s) | 0.98x+0.22 | 0.988 | 170582 | 0.98x+0.34 | 0.991 | 170587 | 0.999x-0.116 | 0.998 | 171085 |

are measured through the same inlet and thus are directly coincident. The majority of the missing COMA data were during above-plume transit legs (49.1% of the missing data) and/or occurred under conditions of very low humidity outside of the biomass burning plume (62.7% of the missing data) due to the 50 ppmv minimum instrument threshold of COMA. Regardless, the results are substantially similar using any of the water vapor content datasets.

## 3.2 Observed plume-water vapor correlations

### 3.2.1 ORACLES in-plume measurements

Examining the correlation between the biomass burning tracer CO and the water vapor content $q$ within the plume layer (i.e., excluding boundary layer altitudes, here defined as below 2km), we see a consistent pattern of elevated humidity with high CO. Figure 3 shows correlations between CO and $q$ for each individual flight, for all altitudes above 2km (to isolate the plume altitudes from those with boundary layer influence). We find similar results for a variety of spatial, altitudinal, and temporal subsets above the planetary boundary layer; in other words, there does not appear to be a single altitudinal or latitudinal range which dominates this relationship for the dataset as a whole. We note that the results are also similar for aerosol extinction and scattering coefficient (Supplementary Figure S3). The amount of water vapor seen here is consistent with the 5g/kg moisture levels reported by Deaconu et al. (2019) for August and September (compared with 2.5g/kg in June/July), and indeed during ORACLES we frequently see values of 6g/kg or greater in the free troposphere.

Each of the flight days in Figure 3 shows a robust linear correlation, and some of the flights show especially linear correlations between CO and $q$, specifically the flights on 8, 10, 12, and 14 September 2016 (middle row). The first three of these flights were along the routine diagonal covering a fairly significant portion of the SEA extending northward to 10°S, 13.5°S, and 9.5°S, respectively. The flight on 14 September is classified as a radiation flight of opportunity, and while it did not follow the routine path, it still sampled somewhat diagonally from Walvis Bay (out to 16°S and a maximum westward extension of 7.5°E). While the correlations appear as generally stronger for routine flights, most of the flights of opportunity show strong correlations as well ($R^2 > 0.8$; Table 3). The notable exceptions to this are the flights of opportunity on 20 and 24 September





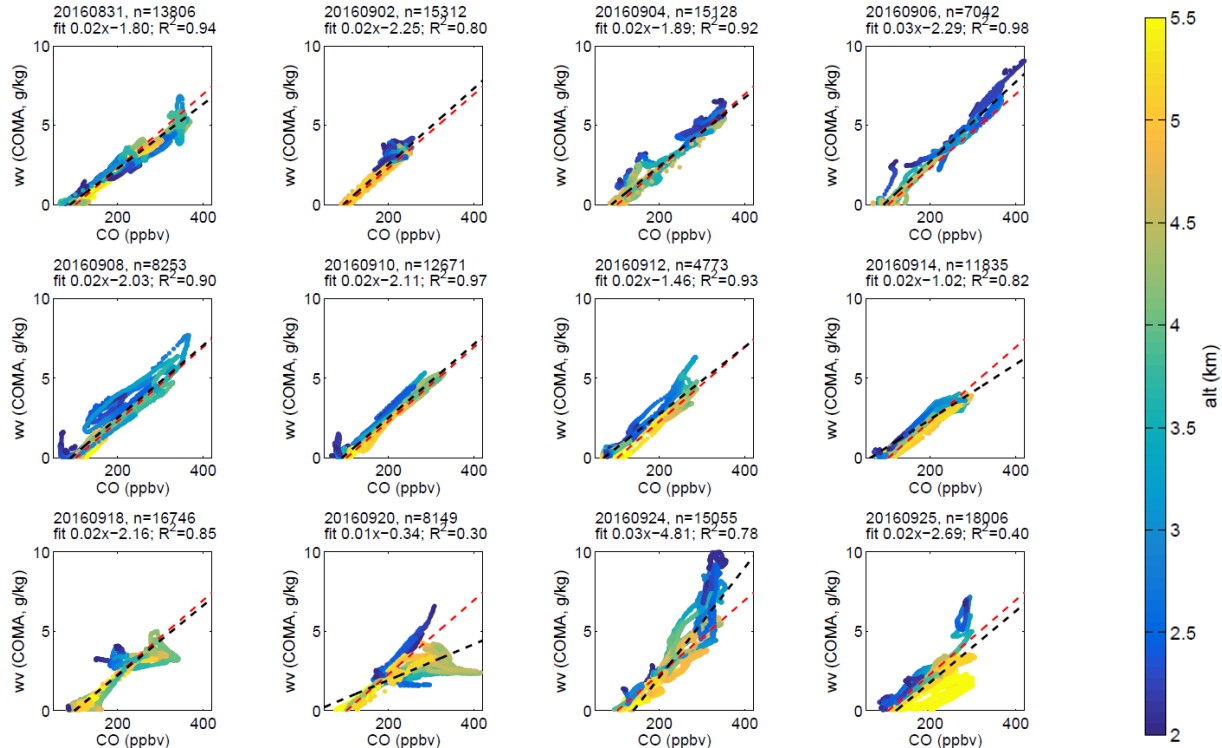

**Figure 3.** ORACLES-2016 specific humidity versus CO, by flight. Here we show only altitudes substantially above the planetary boundary layer (>2km), so as to highlight the correlations at plume level. Black lines show the total least-squares fit to each individual flight ($z > 2$km) and the red line shows the fit through all flights combined. All correlation coefficients are significant to two decimals ($p < 0.01$).

which were both particularly close to the coast; when subdivided spatially over all flights, the relationship is more variable for the more southern coastal areas (Figure 4). On 20 September, dust was also observed during a portion of the flight; this could indicate that the air mass sampled on these days had a different origin and different trajectory upon exiting the continent (i.e., directly easterly), compared with the typical conditions of the elevated biomass burning aerosol layer (i.e., a more northwesterly

5    recirculation from an origin at AEJ-S latitudes). On 24 September, an unusually high boundary layer height was observed with westerlies below 3.5km; this anomalous meteorological condition between 15-20°S may be responsible for the slightly weaker correlation that day. The routine flight on 25 September also has a lower correlation than the other flights. Examining the data shows that there is a shift in the CO-$q$ slope with altitude during this day, which is not seen on other flights; for smaller altitude subsets during this flight, the correlation is stronger. We also note that these flights are the last flights of the deployment, and

10   thus may be capturing an expected seasonal shift compared with the flights earlier in September.

Figure 4 shows a frequency distribution of these same data subdivided spatially, highlighting how remarkably consistent the slope of this relationship is. Humid, smoky air exits the continent at roughly 10°S in the south Atlantic Easterly Jet (AEJ-S), but we see that even at higher latitudes (lower rows), farther from the latitudes of the AEJ-S, the CO-$q$ relationship is strongly





**Table 3.** Correlations between free tropospheric CO and $q$ ($z > 2$km) from in situ instruments as shown in Figure 3, and correlations between AOD and CWV ($z > 1.3$km) from 4STAR as shown in Figure 5, by flight for ORACLES 2016. All correlations are significant to $p < 0.001$. Note that the different altitude limits are due to different methodologies. "Routine" and "opportunity" indicate whether the flights were along the northwest diagonal or near-coast (Section 2.1).

| date | flight | CO vs $q$ 2-6.3km | | AOD vs CWV 1.3-5km | |
|---|---|---|---|---|---|
| | | $R^2$ | # points | $R^2$ | # points |
| 20160831 | routine flight: PRF02 | 0.94 | 13806 | 0.958 | 9159 |
| 20160902 | opportunity flight: PRF03 | 0.80 | 15312 | 0.917 | 14676 |
| 20160904 | routine flight: PRF04 | 0.92 | 15128 | 0.904 | 4136 |
| 20160906 | opportunity flight: PRF05 | 0.98 | 7042 | 0.877 | 7546 |
| 20160908 | routine flight: PRF06 | 0.90 | 8253 | 0.959 | 10288 |
| 20160910 | routine flight: PRF07 | 0.97 | 12671 | 0.977 | 12391 |
| 20160912 | routine flight: PRF08 | 0.93 | 4773 | 0.909 | 6508 |
| 20160914 | opportunity flight: PRF09 | 0.82 | 11835 | 0.957 | 11795 |
| 20160918 | opportunity flight: PRF10 | 0.85 | 16746 | 0.845 | 15882 |
| 20160920 | opportunity flight: PRF11 | 0.30 | 8149 | 0.895 | 6999 |
| 20160924 | opportunity flight: PRF12 | 0.78 | 15055 | 0.855 | 10471 |
| 20160925 | routine flight: PRF13 | 0.40 | 18006 | 0.862 | 9253 |
| all flights | | 0.88 | 146776 | 0.817 | 121984 |

linear, and with much the same range in values down to $\sim 18°$S. The main exception is the coastal Zone 6 (16-18°S and 8-12°E; Figure 1), which is influenced by the observations from 20 September. Overall, this suggests that the range of concentrations observed are present as a given airmass exits the continent, and is not progressively diluted via mixing during transport.

### 3.2.2 Column ORACLES measurements

5 The 4STAR retrievals of AOD and column water vapor (CWV) are measured along the aircraft-to-sun light path and thus represent the full above-aircraft airmass, rather than the values at the aircraft altitude. While some impact of ambient humidity is to be expected due to hygroscopic swelling of aerosols (increasing AOD), it is nonetheless still instructive to examine these parameters as they compare to inlet-based instruments. A prior study of the ORACLES-2016 data (Shinozuka et al., 2020) estimated that the effect of aerosol hygroscopic swelling on extinction was fairly minimal in the free troposphere, with an

10 ambient-RH/dry ratio less than 1.2 for 90% of measurements, suggesting the same may be true for AOD. Figure 5 shows the correlation between the 4STAR AOD at 500nm and the CWV for 1s data from all 2016 flights from above the boundary layer (here, $> 1.3$ km) to upper plume level ($\leq 5$ km). The 4STAR instrument provides a different geometric perspective from that of the inlet-based measurements described above, yet shows similar results, providing additional evidence of the observed linearity between $q$ and smoke concentration. The different altitude ranges compared with Figure 3 are due to the different



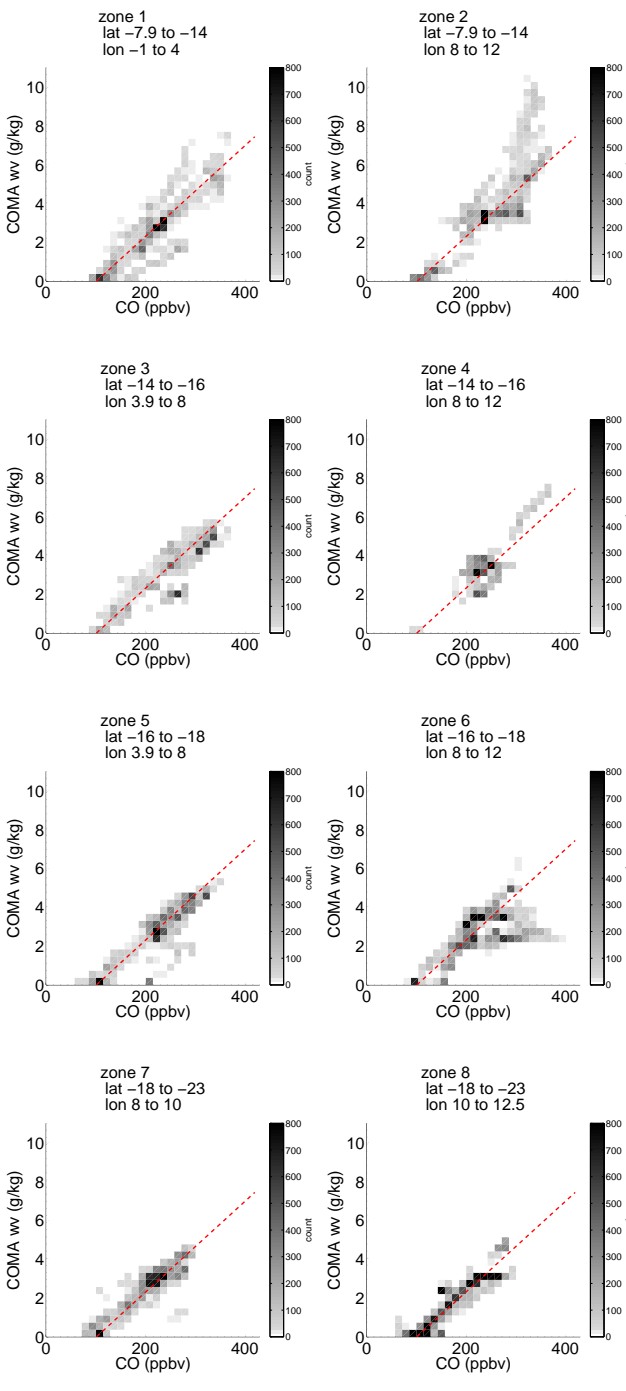

**Figure 4.** Frequency distribution (heat maps) of ORACLES-2016 specific humidity from COMA versus CO for ($z > 2$km), subdivided zonally according to the boxes shown in Figure 1. The red line shows the fit through all 2016 data (as in Figure 3).





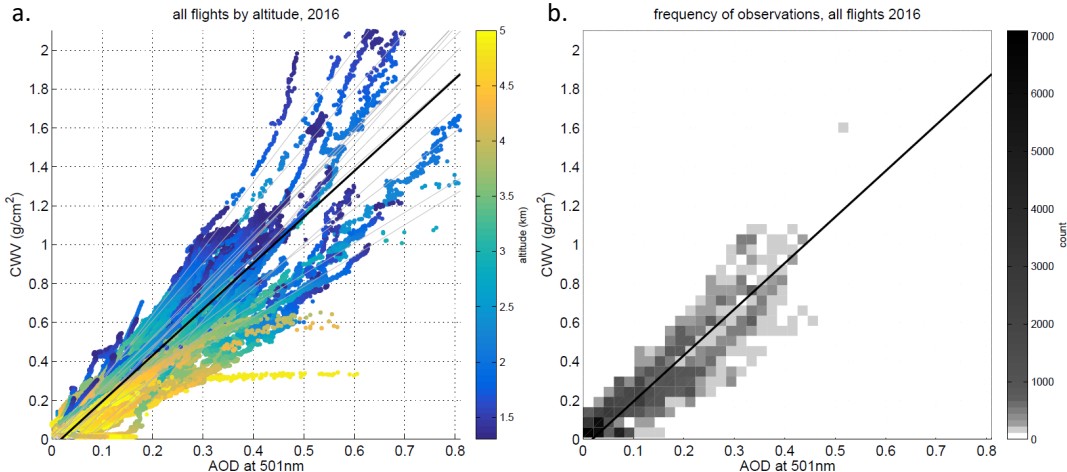

**Figure 5.** 4STAR AOD at 500nm versus column water vapor (CWV) for altitudes above the boundary layer to within the BB plume (1.3 to 5km) **a.** by altitude for all flights and **b.** as a frequency heatmap over all flights. Thin grey lines show the total-least-squares fits through individual flights while the thicker black line shows the fit through all data. The correlation coefficients are fairly high for individual flights (Table 3) and only slightly lower for all flights combined ($R^2 = 0.82$).

instrument requirements and capabilities, i.e., 4STAR observations from within the plume give only partial vertical profiles as 4STAR measures only the airmass above the aircraft at a given time. Thus measurements from below the BB aerosol plume are valid and even preferred for 4STAR, whereas the inlet-based instruments are less useful at lower altitudes when there is a lack of plume loading. The largest range in AOD (and CWV) is seen on 24 September, near the coast, consistent with Figures
5  3 and 4.

The 4STAR observations demonstrate that the plume/vapor relationship is consistent through the plume column and not solely at the instantaneous altitudes and locations as seen by the inlet-based instruments. We note this is also consistent with the results of Adebiyi et al. (2015) who showed that upper-level ($\sim$ 700hPa, roughly 3.2km) humidity from radiosondes corresponded to conditions of high AOD from satellites, albeit this was farther offshore at St Helena Island. The fact that we
10  see a strong linear correlation between markers of the biomass burning plume and atmospheric water vapor from multiple instruments and over multiple flights is a strong indication of the robustness of this relationship over this region during the ORACLES-2016 time period.





### 3.3 Do reanalyses/models capture the relationship seen in the observations?

Having established the robust CO-$q$ relationship over the southeast Atlantic Ocean as seen in these observations during ORACLES-2016, we next seek to explore the larger mechanisms by which this relationship has developed. The source region for ORACLES BB observations includes widespread seasonal grassland savannah fires over central and southern Africa

(e.g., van der Werf et al., 2010; Redemann et al., 2020) and sees little variability in either fuel source or combustion efficiency (e.g., Vakkari et al., 2018). We wish to take a broader perspective which incorporates these continental regions, which is not possible solely using the over-ocean ORACLES aircraft data alone. Thus, we turn to reanalyses and model simulations.

Figure 6 shows the ORACLES flight data from aircraft profiles aggregated and subset to the times and altitudes of the ERA5, ERA-Interim, and MERRA-2 reanalyses, and the WRF-CAM5 and WRF-Chem models, with different reanalysis altitude

ranges distinguished by color and shape. We note that this is a subset of the data shown in Figure 3, but the CO-$q$ relationship shown here is consistent with that of the full dataset. For each of the altitude ranges–boundary layer (square), boundary layer-influenced (triangle), or plume-level (circle)–there is good agreement between ERA5 and the aircraft observations (Figure 6a) from the surface through the plume level. An exception is at altitudes at the top of the boundary layer ( $\sim$570m; squares), where ERA5 often underestimates water vapor, perhaps due to difficulties in determining boundary layer height over the ocean

surface. Despite this, the humidity at surface level agrees well with the observations and, more importantly in the context of this study, the existence, magnitude, and location of elevated water vapor for plume altitudes is also well represented in ERA5. It is reassuring that this newest ECMWF product agrees so well with the aircraft observations ($R^2 = 0.79$ for $z > 2$km), and this gives us confidence that the ERA5 meteorology may be consistent with real-world meteorology over the continental source region as well. Figure 6b and Figure 6c show the comparisons between aircraft-observed $q$ and ERA-Interim and MERRA-2

reanalysis $q$, respectively. Both of these correlations are rather weaker than that for the ERA5 reanalysis ($R^2 = 0.53$ and $0.40$ for ERA-Interim and MERRA-2, respectively), but both still largely capture the presence of an elevated water vapor signal in the altitudes above the boundary layer. However, both these products also often report this high-humidity air as being at a lower altitude than what was observed by the aircraft observations (an example is shown in Figure S2).

Finally, Figure 6d and 6e show the two configurations of WRF described in Sections 2.2.3 and 2.2.4. WRF-Chem $q$ shows

a strong correlation with the observed $q$, in line with that of ERA5 ($R^2 = 0.79$ for both products for all altitudes $> 2$km), which is not surprising due to WRF-Chem's daily initialization with ERA5 reanalysis meteorology. The WRF-CAM5 water vapor is more weakly correlated with the observed water vapor ($R^2 = 0.48$, more in line with the results from MERRA-2 and ERA-Interim). This difference is likely due in part to the different meteorological fields used (NCEP versus ERA5), and also to WRF-CAM5's less frequent initializations (5-day versus 1-day), allowing it to drift farther from the "actual" meteorology and

chemistry conditions between initializations. Given these results alone, one might be discouraged by the possibility of using MERRA-2 or either WRF configuration in this analysis, but this isn't the full story. Although the water vapor co-location is poor, we find that the relationship between CO and $q$ does hold over the flight path (Figure 7). Here, interestingly, the results are flipped: MERRA-2 and WRF-CAM5 show comparatively better correlations between CO and $q$ ($R^2 = 0.56$ and $0.71$, respectively, compared with $R^2 = 0.78$ in the observations), while WRF-Chem now shows more variability in CO-$q$ conditions



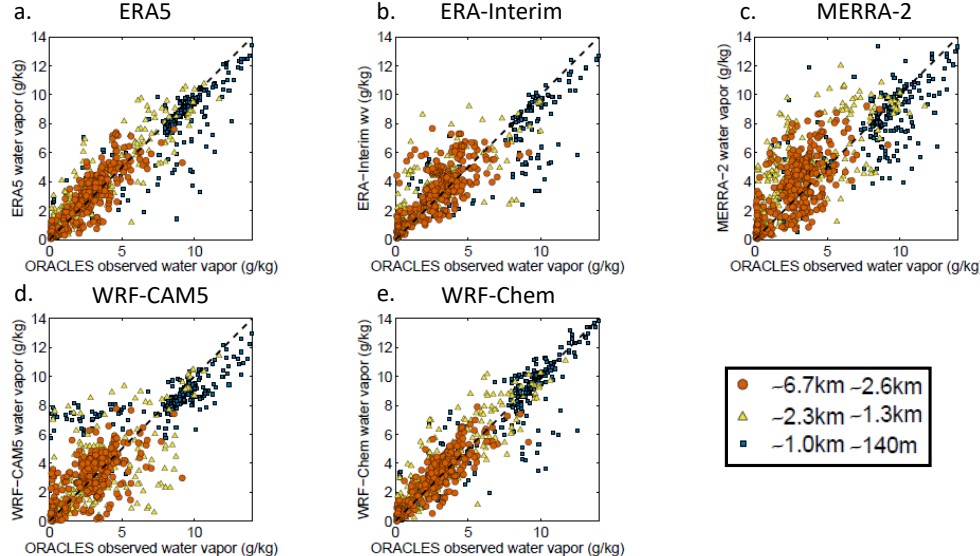

**Figure 6.** ORACLES water vapor measurements compared with reanalyses and models subset to the locations of aircraft profiles and altitudes of ERA5 outputs. Observations are averaged within ±50m of the reanalysis levels; MERRA-2 reanalysis is averaged over the time of the aircraft profile and interpolated to ERA5 altitudes for ease of comparison. Free tropospheric altitudes are shown by circles, smaller squares are the boundary layer, and triangles are intermediate altitudes. Here we see that **a.** the ERA5 water vapor and the observed water vapor subset to ERA5 altitudes show good agreement within the plume layer and for the lower boundary layer ($R^2 = 0.79$ for $z > 2$km); agreement is poorer for **b.** ERA-Interim ($R^2 = 0.53$), **c.** MERRA-2 ($R^2 = 0.40$), and **d.** WRF-CAM5 ($R^2 = 0.48$), although a linear CO-$q$ relationship is still seen. **e.** WRF-Chem initialized from ERA5 shows better agreement ($R^2 = 0.79$).

and thus a poorer correlation between the two ($R^2 = 0.49$). The fact that the CO/$q$ correlation is fairly high for MERRA-2 and WRF-CAM5 even while the observed/modeled $q$ correlation is low essentially indicates that while these two products aren't placing a given airmass exactly where and when it is observed by the aircraft, the consistent relationship between the plume and water vapor is maintained in the alternate location. We must also consider the differences in model emissions and

5    meteorological configurations to potentially explain this. MERRA-2 and both WRF models use QFED emissions, albeit with different implementation in each. Because WRF-CAM5 has the best correlation between CO and $q$, and the longest independent run length, it seems plausible that the periodic reinitialization of each model's meteorology independent of its emissions weakens the correlation between the two. This would be because the reinitialization will "correct" the meteorology (water vapor) towards the reanalysis, while the chemistry (CO) will be adjusted independent of the meteorology, and to a different

10   degree. This would explain why the 3-day runs (5-day minus 2-day spin-up) of WRF-CAM5 show a stronger correlation than WRF-Chem (with daily reinitialization) or the MERRA-2 reanalysis. We also note that MERRA-2 and WRF-CAM5 report lower CO for higher water vapor (i.e., the slope between the two variables is steeper than in the observations) whereas the opposite is true for WRF-Chem. Overall, this pattern suggests that the CO-$q$ relationship is sustained through dynamics affecting both properties equally, i.e., not diabatic processes such as cloud formation which could decrease the water vapor,





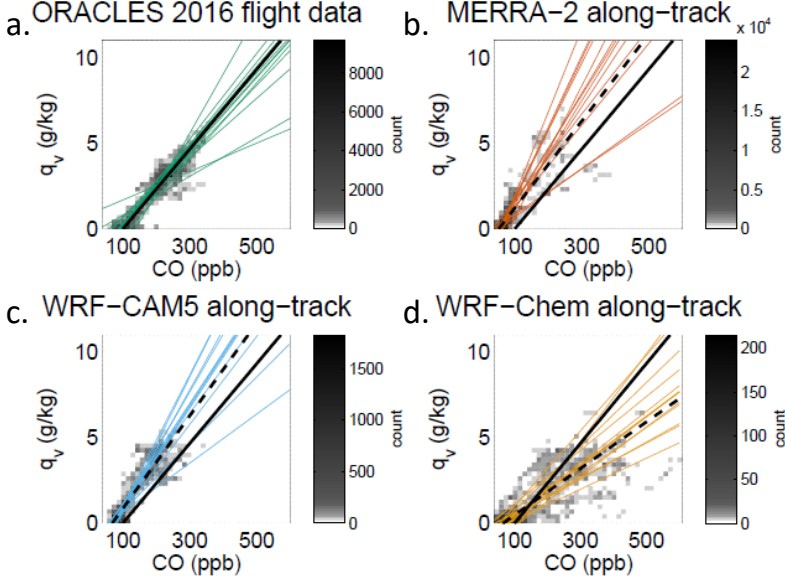

**Figure 7.** CO vs $q$ from **a.** aircraft observations ($R^2 = 0.78$), **b.** MERRA-2 reanalysis ($R^2 = 0.56$), **c.** WRF-CAM5 ($R^2 = 0.71$), and **d.** WRF-Chem ($R^2 = 0.49$) for all flights, for altitudes $z > 2$km. Individual colored lines show the total least-squares fits for individual flights, and the black lines show the averages of all observed flights (solid) compared to each model (dashed).

and moreover that this is also true within the considered models. Given this context, we conclude that, while not perfect, the different strengths (and limitations) of each of these models may be useful in understanding the mechanisms involved in the real world.

Figure 8 shows vertical profiles of water vapor from COMA subdivided spatially by latitude and longitude grids according to the boxes shown in Figure 1 (the same divisions used in Figure 4), with routine flight paths in the left column and coastal flights on the right. Each subplot shows profiles of the nearest co-located ERA5 reanalysis points, for comparison. This spatial division by aircraft profile highlights both the consistency in the vertical structure of the plume observed by aircraft and shown by ERA5, and the differences in this vertical structure in different regions of the SEA. In terms of the spatial differences, Zone 2 (top right) has consistently the highest measured water vapor (4-11 g/kg) and CO, possibly due to its proximity to the location of the AEJ-S ($\sim 10°$S). Also, along the routine diagonal (i.e., farther from the coast), we more frequently see a dry/clean gap between the humid plume and the more humid boundary layer, plus a greater plume strength compared with the near-coast regions at the same latitude (see also Figure 4). In contrast, the more coastal flights often see either more humid, higher-CO air masses at lower altitudes, or constant CO and $q$ at all altitudes (Figure 4). Finally, we note that Figure 8 shows again how consistently well the ERA5 reanalysis performs when compared to the aircraft observations, even in the case of varying profile type. There is a good deal of variability in this structure in different latitude/longitude ranges (e.g., high- and low-altitude plumes with substantial vertical variation or a fairly consistent magnitude with altitude) but these differences are consistent between both ERA5 and the observations.





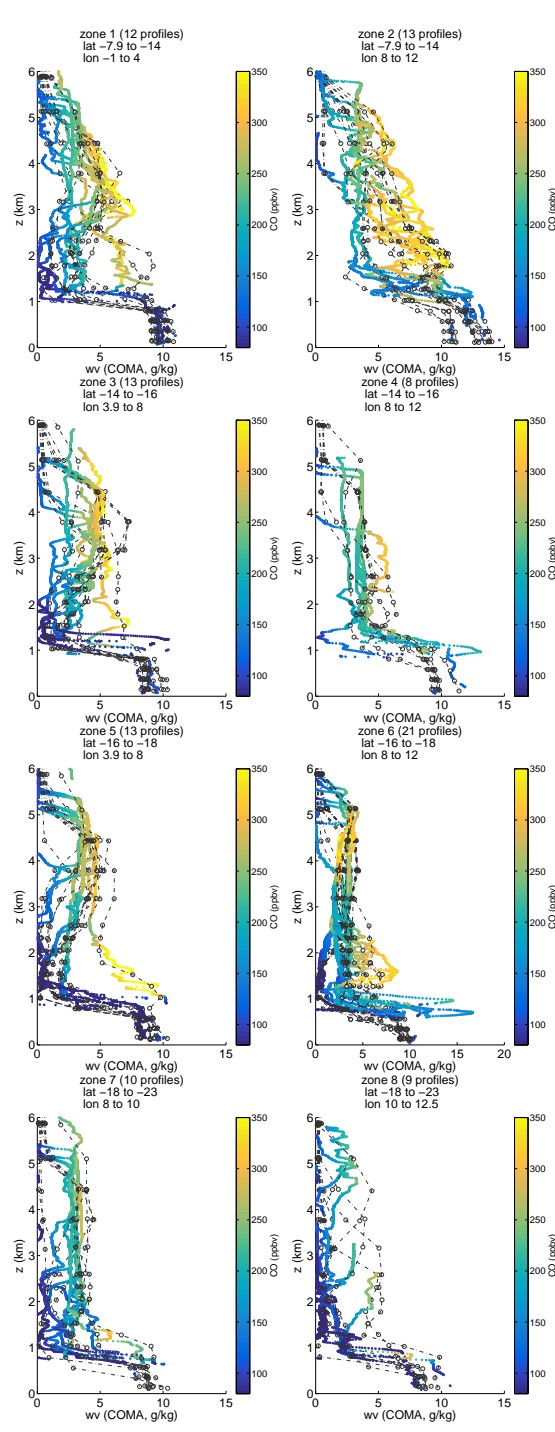

**Figure 8.** Profiles of specific water vapor measured by COMA in ORACLES-2016 (solid lines), subdivided by spatial location. Colors indicate the CO concentration from COMA. Dot-dashed lines show spatiotemporally co-located ERA5 reanalysis profiles for each aircraft profile, which captures the variability in vertical structure reasonably well.



### 3.4 The larger scale perspective: continental origins of the linear relationship

Our results thus far are consistent with previous satellite- and reanalysis-based work which described both the same pattern of elevated water vapor coinciding with biomass burning aerosols over different parts of the SEA (e.g., Adebiyi et al., 2015; Deaconu et al., 2019), and the importance of the south African easterly jet (AEJ-S) in transporting continental airmasses over the southeast Atlantic Ocean (e.g., Adebiyi and Zuidema, 2016). Having shown that several models and reanalyses are able, to some degree, to capture the presence of an upper-level water vapor signal during ORACLES-2016, in this section we focus on the reanalyses to gain more insight into the origins of this pattern over the biomass burning source region. Specifically, we may reasonably expect that due to the excellent agreement between the ERA5 reanalysis and the observations in the ORACLES SEA sampling region, ERA5 may give an accurate picture of meteorological context for the airmass origin over the continent and its evolution during its westward transport. MERRA-2, while not as directly translatable to aircraft measurements, may yet allow us to complete the picture by showing how $q$ relates to CO concentration.

Figure 9a shows a Hovmoller timeseries of ERA5 atmospheric water vapor with longitude at 600hPa ($\sim 4.4$km; identified by Adebiyi and Zuidema (2016) as the altitude of max AEJ-S strength), averaged over 7.75°S-14°S. These latitudes are chosen to encompass the usual range of the AEJ-S while overlapping with the upper extent of the ORACLES flight data (Zones 1 and 2 in Figure 1). These $q$ contours are overlaid with average horizontal wind vectors at the same altitude. A few features are obvious from this reanalysis: first, multi-day episodes of high water vapor conditions are seen to originate over the continent and are advected westward when zonal wind speeds are high. That is, an elevated water vapor signal is frequently present up to 5km over the continent and these humid airmasses are transported in the easterly jet only under conditions of high zonal wind speeds. Second, we note that there is a notable diurnal cycle in $q$ over the continent, likely driven by the diurnal cycle in the continental boundary layer development. The timing of the diurnal maximum $q$ varies substantially with altitude (as will be discussed shortly). While Figure 9 shows the 600hPa pressure level, the results are largely the same for pressure altitudes 700-500hPa (i.e., the range of the AEJ-S; Adebiyi and Zuidema, 2016), and for latitude subsets within this range. For more southern latitudes, the reanalysis shows much weaker zonal winds, less water vapor at higher altitudes, and no direct connection between continental and over-ocean conditions at the same latitude; the direct east-west transport is not observed. While the AEJ-S ranges from 5-15°S, between $\sim 5$ and 8°S there is likely a combination of dry and moist convection present, whereas dry convection is likely to dominate south of 10°S. Either type of convection will result in elevated $q$ at the AEJ-S altitudes. This pattern of transport is consistent with the BB source region being at more equatorial latitudes even for the more southern ORACLES observations, i.e., recirculation of smoky, humid air from the north to the south, as was also shown by Adebiyi and Zuidema (2016). The broader meteorological features were discussed in more detail in Redemann et al. (2020).

A similar pattern is seen in CO reported by MERRA-2 (Figure 9b): periodic events of westward CO transport are co-located with water vapor transport events, driven by the zonal winds. Both the zonal winds and water vapor are generally similar between the MERRA-2 and ERA5 reanalysis. The water vapor and CO are qualitatively similar in the WRF models as well, although we observe a distinct discontinuity in the timeseries of these models which corresponds to the (daily for WRF-Chem, or 3-daily for WRF-CAM5; Figure S4) reinitialization. This lends credence to the idea that model reinitialization may be





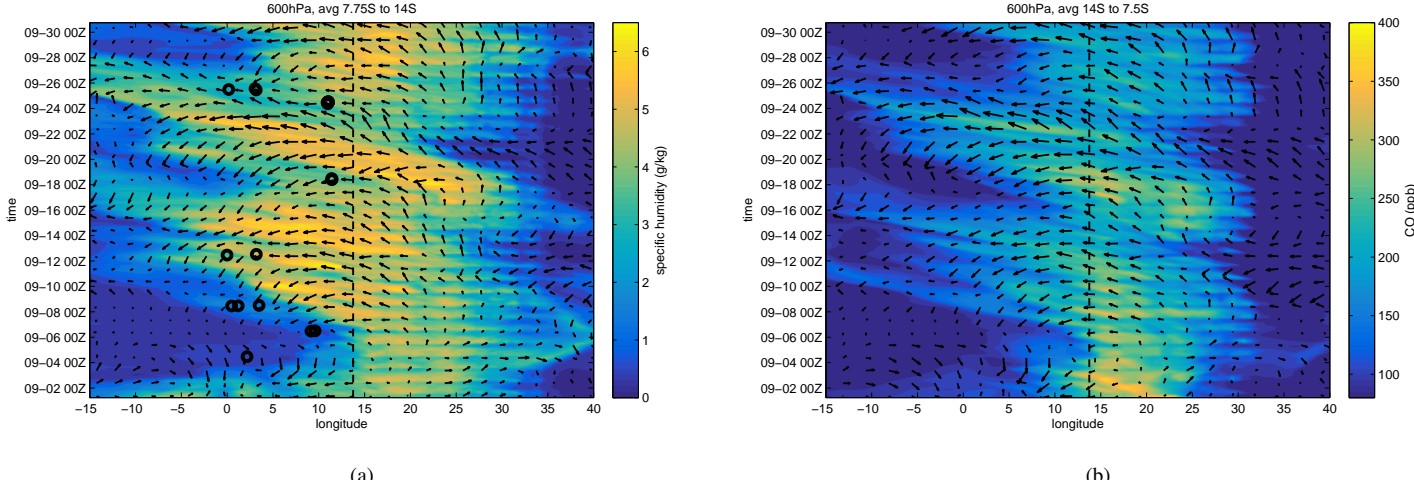

|     |     |
| :-: | :-: |
| (a) | (b) |

**Figure 9.** Hovmoller plot showing the development and transport of airmasses from continental Africa over the SEA, with colors showing **a.** water vapor from ERA-5, and **b.** carbon monoxide from MERRA-2, at 600hPa (∼4.4km), based on 3-hourly timesteps. The location of the African shoreline in this region is indicated by the black dashed line, and data are averaged between 7.75° and 14°S (note the domain is slightly larger for MERRA-2 due to the model resolution). Black circles show the locations and times of ORACLES aircraft profiles within this region. In the east (right-hand) of each plot, the diurnal convection cycle is evident, showing increased water vapor (CO) at this altitude during the daytime; in the west (left-hand), episodes of water vapor (CO) are seen as these continental airmasses are advected by the AEJ-S. Wind vectors do not scale between the two panels, although the patterns are seen to be largely similar between the two reanalyses.

responsible for the weaker correlations in these products (WRF-Chem in particular; Figure 7), as $q$ and CO are adjusted to differing degrees during this process. The fact that the correlations persist between reinitializations but then are lost again suggests that any removal/mixing processes over the SEA Ocean are affecting CO and $q$ equally; i.e., the air is not subject to significant diabatic processes or cloud formation during transport, which could lower $q$ without affecting CO.

5  Figure 10 shows a timeseries of the vertical profiles of ERA5 humidity, over the same latitude range as Fig. 9, averaged over two distinct longitude ranges (lavender boxes in Figure 1): the eastern continental source region (Figure 10b, 15 to 20°E) and the western ORACLES region (Figure 10a, 7.5 to 12.5°E). Selected zonal wind speed values are overlaid as black contours (the thickest line shows the 6 m/s easterly zonal velocity threshold for the AEJ-S (Adebiyi and Zuidema, 2016), with thinner lines showing 8 and 10m/s). With the exception of early in the month, when a baroclinic disturbance was present to the south, the

10  AEJ-S is seen to be almost always present and centered around 4km (∼ 650hPa) in both these domains, though the jet varies in magnitude both on a diurnal cycle and throughout the month. Although there are multi-day humidity (and AEJ-S) episodes, over the continental source region there is a strong diurnal variation in both zonal wind speed and water vapor content which is dampened once the jet exits the continent.

Figure 11 shows the MERRA-2 winds and CO over the same two regions. The pattern is similar: MERRA-2 also shows

15  the frequent presence of the zonal jet, with a strong diurnal cycle in wind speed over land, and the CO values again indicate





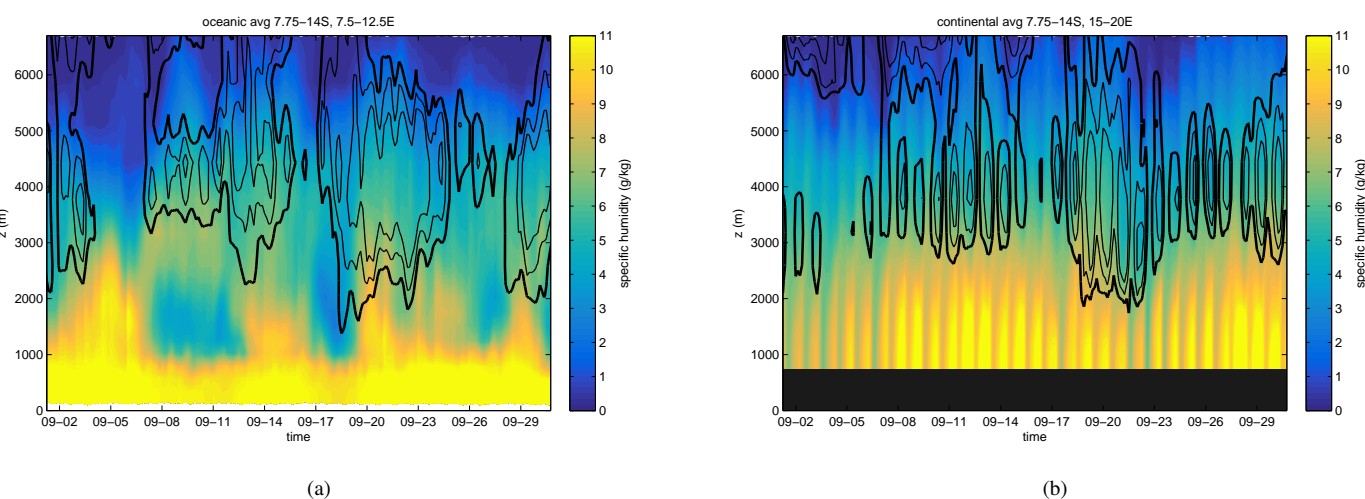

**Figure 10.** Oceanic **(a)** and continental **(b)** specific humidity (shaded) overlaid with zonal wind speed (black contours) from the ERA5 reanalysis. The thick black lines indicate the threshold of the AEJ-S (6 m/s) with thin black lines showing 8 and 10 m/s easterly velocities.

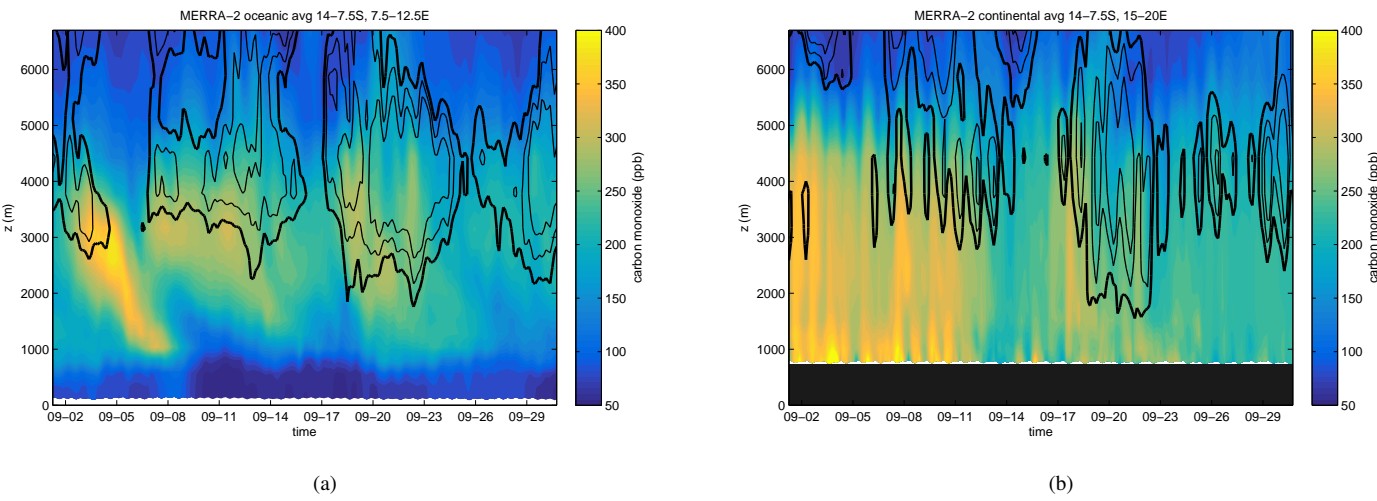

**Figure 11.** Oceanic **(a)** and continental **(b)** CO (shaded) overlaid with zonal wind speed (black contours) from the MERRA-2 reanalysis. The thick black lines indicate the threshold of the AEJ-S (6 m/s) with thin lines showing 8 and 10 m/s easterly velocities.





boundary layer influence reaching to above 5km, propagating upward in time. While the presence of the AEJ-S over the SEA corresponds to significant carbon monoxide, we also see how this high-CO airmass may disperse out into the broader region (e.g., the episode starting around 4 September at 3km over the ocean region is transported down to 1km by 6 September in the absence of the strong zonal winds). The direct comparison between MERRA-2 profiles and aircraft observations suggested a

potentially too-strong subsidence, resulting in a lower-altitude $q$ maximum (Figures 6, S2); indeed, Das et al. (2017) previously documented a subsidence in MERRA-2 which was greater than that inferred from satellite observations. For this particular instance there was a sustained downward motion at 700hPa in both ERA5 and MERRA-2 between 4-6 September, which may be responsible for this episode seen in both reanalyses (Figures 10a, 11a). Regardless, even in a case of too-strong subsidence in MERRA-2, this issue itself will not affect the relationship between CO and $q$ once it's over the SEA, but rather just its

location. It is clear from the two reanalyses that continentally-influenced air over the SEA remains for a sustained period of time and is transported both horizontally and vertically throughout the region while retaining high-$q$ and high-CO amounts.

     Further insight can be gained by examining the diurnal cycle directly at individual pressure levels. Figure 12 shows time series of key meteorological parameters: zonal winds, water vapor, pressure vertical velocity, and potential temperature ($u$, $q$, $\omega$, and $\theta$, respectively) from ERA5, and the same parameters plus CO from MERRA-2, at constant pressure levels of 550 and

650hPa (approximately 5.1 and 3.7km; just above and below the AEJ-S maximum). The bottom panels of Figure 12 show the diurnal cycles of each day normalized to scale between a unitless 0 and 1, and then averaged over all days in September 2016. While this doesn't provide any information on the magnitude (this is captured in the panels above), it does illustrate the relative timing of the minima and maxima of each variable through the diurnal cycle, as well as providing a qualitative idea of the strength of this diurnal cycle throughout the month (i.e., when the maximum in an average curve approaches 1 as $u_{650\text{hPa}}$ at

15Z, this is an indication that wind speed consistently peaks at that time each day, and in contrast, the flatter curve of $CO_{650\text{hPa}}$ shows that the diurnal cycle either does not vary throughout the day, or peaks at different times on different days; from the above panel for CO, we can see in this case it's the former). Taken together with the upper panels, this visualization allows us to examine the the strength of the diurnal variations compared with multi-day events, how each of these parameters at a given altitude is offset from the others at the same height, and thus the range of airmass conditions which exit the continent in the

AEJ-S.

     In the previous figures, we saw a daily upward propagation in the continental water vapor (Figure 10b) and the similar feature in CO (Figure 11b), likely due to diurnal heating causing daytime boundary layer growth over the land. This convection allows the surface air to mix upward and reach strikingly high altitudes ($\sim 5$km) during the day, but the vertical motion is influenced by upper-level subsidence at night. In Figure 12, we note that this pattern propagates upward with a delay: while

daily maximum humidity at 750hPa ($\sim 2.6$km) was generally around 9-12Z, the maximum at 650hPa varies between 12-18Z, and at 550hPa it is still later, between 15-21Z. Again we note there is both daily variation and multi-day episodes, which both vary with altitude. Specifically, the diurnal variability in $q$ is strong at both 650 and 550 hPa, whereas for CO, there is a distinctly stronger diurnal cycle at 550 hPa; the reverse is true for $u$, which has larger daily variation at 650 hPa. The diurnal cycle also varies throughout the month, with a somewhat weaker diurnal cycle in both CO and $q$ when the zonal winds are

strongest (e.g., 19-21 September).





(a)                                                                                              (b)

**Figure 12.** Time series of (top to bottom) zonal winds ($u$, m/s), specific humidity ($q$, g/kg), CO (ppb), potential temperature ($\theta$, $^\circ$C), and vertical velocity ($\omega$, Pa/s), at 650hPa (left) and 550hPa (right) for MERRA-2 (colored lines) and ERA5 (black lines) reanalyses. Each parameter has a distinct diurnal cycle except CO at 650hPa. The 650 and 550hPa panels for a given parameter are on the same scale so as to highlight differences in diurnal cycle magnitudes with altitude, though shifted to capture the full range at each level. Shading indicates night (6pm-6am). The horizontal dashed line in the $u$ panel shows the 6m/s AEJ-S wind speed threshhold (Adebiyi and Zuidema, 2016), and the horizontal dashed line in the $\omega$ panel shows the 0 Pa/s threshhold which separates rising ($-\omega$) from sinking ($+\omega$) vertical motion. Note that easterly $u$-winds are given by negative values. The bottom panel shows the composite diurnal cycle for each variable from MERRA-2 (solid) and ERA5 (dashed) overlaid on one another (colors the same as above), normalized to a diurnal minimum of 0 and maximum of 1, and then averaged over all September days.





We note that while the water vapor over the African continent shows a strong diurnal cycle due to solar heating, the fire strength also has a diurnal cycle following the anthropogenic burning patterns (Roberts et al., 2009). While these timings vary based on location, they generally peak in the late afternoon and are almost entirely extinguished by nightfall (Roberts et al., 2009), which is fairly similar to the timing of daily evaporation and convection over the continent. As mentioned earlier, in

this region, the fire characteristics themselves are fairly consistent over this period (fuel type, combustion efficiency, and burn condition). While the multi-day CO variation does not closely track with that in $q$, the timing of the peaks for an individual day is largely consistent with one another at both levels (minima at 09Z, maxima between 15-18Z; Figure 12 bottom row). CO at the lower altitude varies substantially over the course of several days ($\sim 100$ppbv), the 550hPa CO consistently varies by 50-100ppb within a 24-hour cycle, with the maximum CO between 18Z-00Z, suggesting frequent influence from dry, clean air

above. This suggests that the 550hPa level is influenced by upper level subsidence and mixing on a daily basis, whereas the values at 650hPa are mostly affected by transport in the AEJ-S.

Another piece to the puzzle is the dynamics. Daytime vertical motion over the continent is dominated by solar heating and subsequent convection, as is seen in the substantial daytime increase in potential temperature and the upward propagation of both humid and high-CO air. Overnight, convection is reduced and (when the AEJ-S is active) the zonal wind generally

increases, advecting this air to the west. During times of weak-to-no AEJ-S (e.g., first week of September 2016), the decreasing $q$ and CO overnight at 550 hPa is accompanied by frequent strong subsidence and increasing $\theta$ (due to the subsidence from above in the absence of solar heating), which suggests increased stratification which would inhibit vertical mixing. The vertical velocities in Figure 12 show more frequent subsidence ($+\omega$) at 550hPa versus 650hPa, and $\omega$ at both levels has a maximum (downward velocity) in the early morning (06Z) and a minimum (upward motion) in the late afternoon (15-18Z), which is

consistent with convection caused by diurnal heating. In contrast, during times of strong jet activity (e.g., 18-22 September), the jet still largely strengthens overnight, $q$ and CO decrease, but potential temperature also decreases. Since CO and $q$ still generally decrease during this time, this may indicate that increased shear mixing is happening when the jet is strong, which decreases the CO and $q$ values by mixing the more humid and smoky continentally-influenced air with dry, clean upper-level air. When AEJ-S conditions are weak, and when the potential temperature is relatively high, large-scale subsidence dominates

and stabilizes the atmosphere without much mixing at this interface.

This distinction between high-jet and low-jet conditions is corroborated by Figure 13. This figure shows the CO-$q$ correlations from the MERRA-2 reanalysis along one longitude line over each of (right) the continental source region and (left) the oceanic ORACLES sampling region for the surface-influenced altitudes and for the free-troposphere, respectively. For all data throughout the continental boundary layer over land (top right), the relationship is not as coherent as that observed during

ORACLES, and at individual altitude levels below 550hPa the linear relationship is nonexistent (Figure S5); the low-CO, low-$q$ data are almost entirely driven by the higher altitudes ($>600$hPa). In Figure 13, there is also a frequent condition of (relatively) high-$q$ ($\sim 12$g/kg) and low-CO ($< 300$ppb) which does not correspond to any particular altitude level. In other words, this humid air with a wide range of CO values is frequently present at AEJ-S altitudes, rather than being confined closer to the surface (Figure S5), yet was not observed during ORACLES. At the same time, the linear relationship is seen over the SEA Ocean

for these same latitudes (Figure 13, top left); this is puzzling, since based on our previous analysis (e.g., Figure 9), we expect




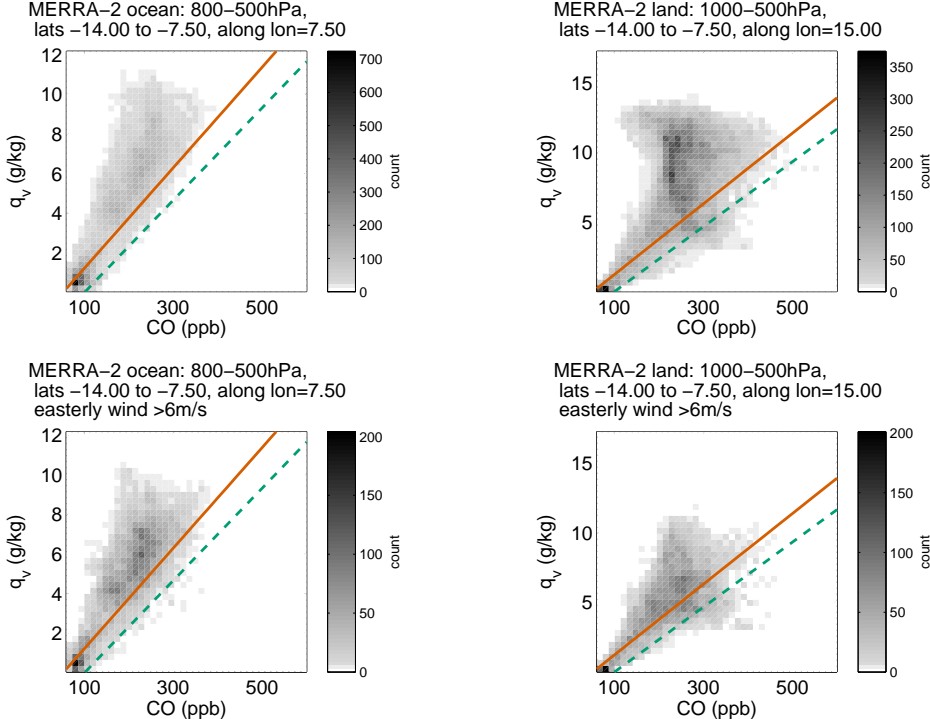

**Figure 13.** MERRA-2 CO and water vapor over land and over ocean, for (top) all observations along 15°E and 7.5°E, and (bottom) only observations for which easterly wind speed was >6m/s. For orientation with previous results, the green dashed line is the fit through all ORACLES-2016 free troposphere flight data and the solid red line shows the MERRA-2 fit coincident with the aircraft observations (both as in Figure 7). A vertically-resolved version of this plot is shown in Figure S5. The results are largely similar for WRF-CAM5 and WRF-Chem (Figure S6).

the eastern continental region to be the direct source for the western oceanic region. When we consider only the conditions of strong easterly transport (Figure 13, bottom), the situation becomes clearer: now, the CO-$q$ relationship over the continent is much closer to the linear relationship observed over the ORACLES region, and over the ocean it is largely similar. Similar patterns are seen in both WRF configurations (Figure S6), with a stronger high-$q$, low-CO feature, likely due to differences in

5  biomass burning implementation between each model.

It is notable that if we consider the CO-$q$ relationship of Figure 13b only for one jet level (e.g., the jet maximum of 600hPa), there is no obvious linear CO-$q$ relationship at all over land (Figure S5). Only starting at the 550hPa level does a linear relationship begin to emerge primarily driven by low-$q$, low-CO conditions. These higher altitudes are at times alternately influenced by both clean, dry upper troposphere air and by humid, smoky surface-influenced air (Figure 12). According to

10  MERRA-2, these values decrease in altitude (as expected) from 5 to 12 g/kg in $q$ and 200 to 500 ppb in CO at 700hPa, to 0 to 5





g/kg in $q$ and 60 to 300ppb in CO at 500hPa. While the maximum $q$ continues to decrease above 500hPa, dropping to 1 g/kg at 400hPa, even at this high altitude the CO doesn't fall below 60ppb. While this may be due to the emissions schema used rather than physical reasons, this is nonetheless consistent with the minimum CO observed by aircraft during ORACLES, suggesting accurate background CO is used by the models.

It thus seems plausible that the mixing between surface and upper troposphere air is occurring over the continent, resulting in a vertical gradient from the surface up through the altitudes of the AEJ-S. Due to the frequent upward convection along with diurnal variations in potential temperature and in zonal winds, airmasses with a specific range of co-associated conditions are selected by the AEJ-S, thus effectively converting these vertical gradients into horizontal gradients over the SEA. Thus the mixing which occurs over the continent and the resulting airmasses are transported over the SEA having this range of

properties, which results in the same linear pattern being present over the broader SEA region. The linear relationship observed during ORACLES is the result of air which left the continent as multiple different levels within the AEJ-S range, and which were subjected to the AEJ-S conditions.

### 3.5    Results from the 2017 and 2018 deployments

As the ORACLES-2016 data represent only about one third of the data collected during ORACLES, we wish to briefly discuss

the context of the latter two ORACLES deployments. As discussed in Section 1, the ORACLES-2017 and -2018 deployments differed from ORACLES-2016 in several key ways. Each deployment occurred, by design, during a different month (August and October in 2017 and 2018, respectively), and thus saw different climatology. The spatial sampling was also significantly different in the two later years (i.e., more northerly; Figure 14) due to the moving of the deployment base to São Tomé. Even between the two latter years, the 2018 flights were generally closer to the continent, whereas the 2017 flights included a

series of flights to, around, and from Ascension Island at 14.4°W (this runway was not available in 2018). Sampling the more equatorial air masses in 2017 and 2018 means these flights sampled more humid air and a deeper boundary layer (Figure 15) even after accounting for the expected seasonal climatological changes. As the biomass burning season peaks in September and shifts geographically through the season, the plume itself, and the prevailing meteorology, would have been different even if the flights had occurred from the same base in all three years (Redemann et al., 2020). Aside from this, the ORACLES analysis

found that there was significant interannnual variability from year to year such that some years saw a peak in BB in September and some saw the peak in August. A more detailed discussion of the broader meteorological and aerosol contexts may be found in Redemann et al. (2020).

Figure 15 shows the CO-$q$ relationship above 2km for a subset of 2017 and 2018 flights. A few key differences are evident between Figures 3 and 15. The most prominent difference is that while the two values are still largely correlated, the near-

universal linearity between CO and water vapor observed in 2016 is largely absent in 2017 and 2018 (grey + colored points). However, when only considering observations within the same spatial range as 2016 (south of 7.75°S and east of 0°E; colored points), the correlations are stronger. We note the total-least-squares fits through the full dataset (blue dashed lines) versus 2016 overlap (lavender dashed lines) are not significantly different for each year, likely due to the dynamic range in CO and $q$ values in both divisions. The more equatorial observations (grey points) are frequently high-humidity/low-CO/lower-altitude

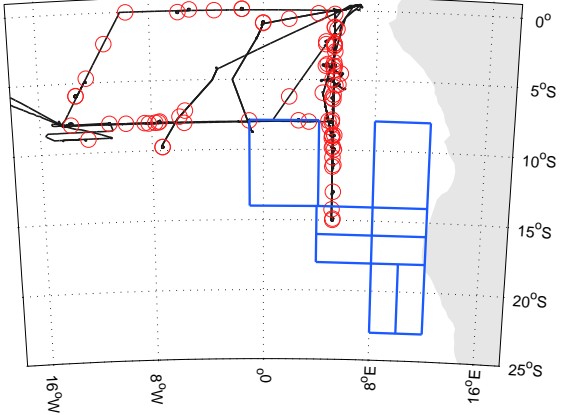 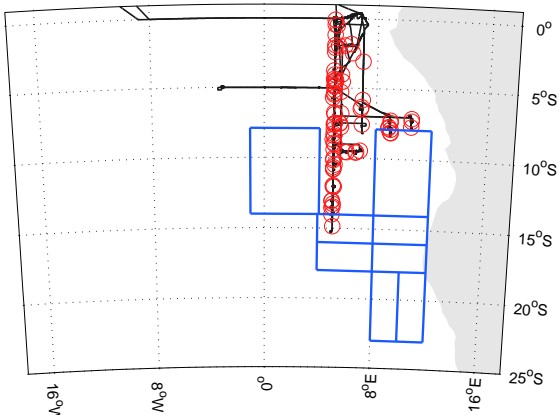

**Figure 14.** Map showing the flight tracks of (left) the 12 science flights + 2 transit flights by the P-3 in ORACLES-2017 and (right) the 13 science flights + 2 transit flights by the P-3 in ORACLES-2018. The blue boxes give the regional subsets used in Figure 1 and Section 3.3, which highlights the difference in spatial sampling between 2017-18 and 2016. While few flights in these years fell within these boxes, quite a few (including the routine flight path) were within the 7.75 to 14°S latitude range discussed in Figures 9 and 10. Note that while the "routine flights" in 2016 followed a SE-to-NW diagonal, the "routine" flight path in the two later years was N-S along 5°E.

observations, particularly in October 2018, indicating boundary layer influence extends to a higher altitude than in 2016. The differences between the three deployments is likely due to the anticyclonic atmospheric circulation at AEJ-S latitudes towards the south. In other words, seasonal variation aside, the 2016 deployment simply sampled more airmasses which were influenced primarily by the BB plume, rather than other more northerly origins of the latter two years. August 2017 more frequently saw
5    higher-CO airmasses with relatively lower water vapor compared with the other two deployments. August climatologically sees more northern convection (compared to that in September and October, when the convection migrates south with the end of winter) and also has a much weaker AEJ-S; the AEJ-S was especially weak in 2017 (Redemann et al., 2020), which may also be a factor in the weaker correlations during this deployment. Of the three years, the correlation coefficients between the two measurements are highest in 2016.
10    The weaker correlations and more humid conditions are thus likely caused by a combination of upward mixing of the oceanic boundary layer in the latter years, the seasonal change in biomass burning sources, and the more equatorial meteorology sampled in 2017 and 2018.





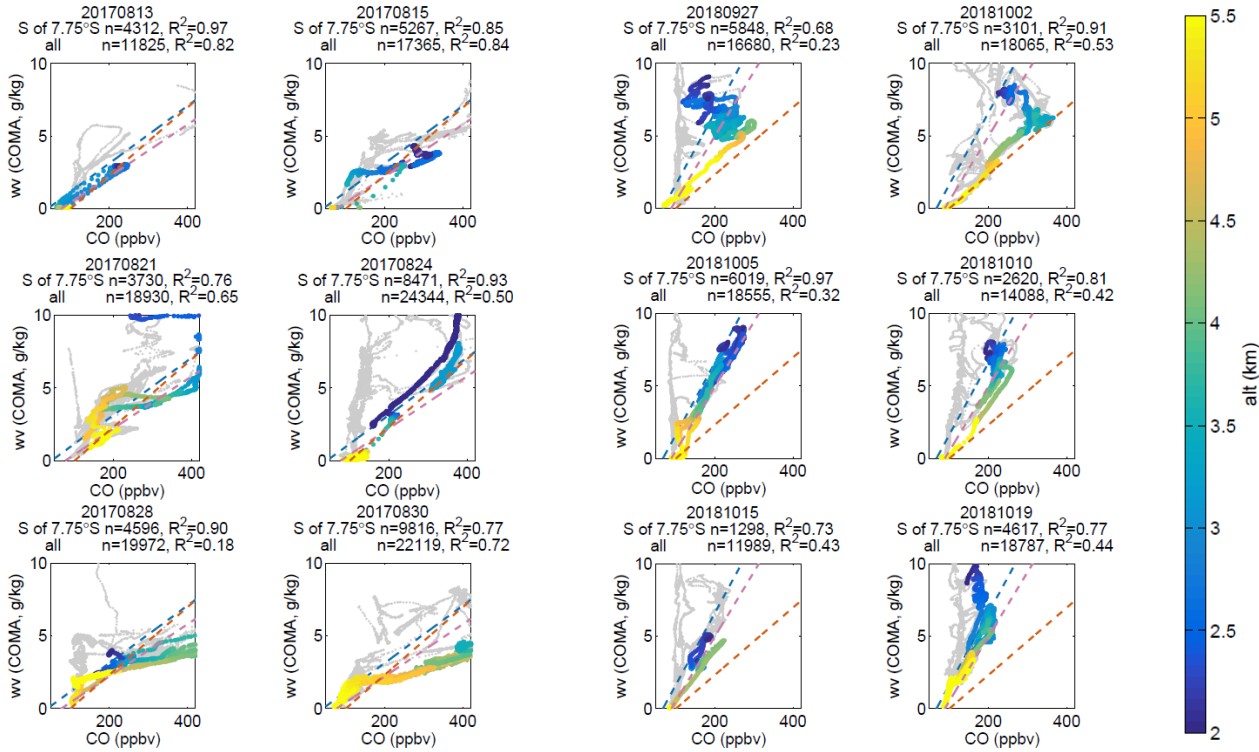

**Figure 15.** ORACLES-2017 (left) and ORACLES-2018 (right) water vapor vs CO, for selected flights, for the subset of latitudes overlapping with the 2016 sampling region (south of 7.5°S; colored), and for the remainder of data at all latitudes above 2km (grey). The thick blue lines show the fits through all 2017 (2018) flights, and the purple lines indicate the fits through the portions of the 2017 (2018) flights which are south of 7.5°S and east of 0°E. The thick red dotted lines show the fits through all 2016 flights as in Figure 3. The variation in this relationship from year to year is evident.

## 4 Discussion

Thus far, we have established that (1) there is a robust linear correlation between water vapor and BB plume strength as measured from several distinct aircraft instruments; (2) this elevated water vapor feature appears, with varying fidelity, in both meteorological reanalyses and free-running climate models; (3) there is frequent deep boundary layer daytime convection over the continent which causes humid/smoky air to be lofted to the altitude of the AEJ-S, which transports it westward; and (4) the linear CO-$q$ relationship is seen over the continent, but only concurrent with a strong AEJ-S condition. We now attempt to synthesize these findings to paint a coherent picture of the evolution of this condition between its source on the African continent and its observation with the ORACLES aircraft. Then, we will briefly explore whether the high water vapor content may be due to some characteristic of the biomass burning itself, or due to some other cause.





## 4.1 Trajectories from emission to observation

Figure 16 shows the example of an ORACLES aircraft profile (ramp) from 10 September 2016 at approximately 10Z (09:58:50-10:10:33 UTC) centered at $15.6°$S, $5.6°$E (south of the AEJ-S range; Zone 3 in Figure 1). We choose this profile as, first, it showed multiple plume layers of varying strength: a main plume layer starts around 3.5km, strengthens to 4km, and continues

above the aircraft range ($\sim$4.2km in this case), with a secondary peak in CO and $q$ around 2.4 km, and a layer of low-CO/low-$q$ between the two ($\sim$2.8 to 3.2 km). Below the second plume layer, there is a gap of much cleaner air around 1.5km, just above the boundary layer. The second reason we choose this profile is that since the ERA5 reanalysis captures these features fairly well at this time and place, including the smaller secondary $q$ below 3km. (We note that MERRA-2 shows this feature as well (purple dashed line), although the main plume layer is too low in altitude compared with the observations).

Next, the map in Figure 16 (top left-center) shows HYSPLIT back trajectories from three locations within this profile: 4km (the maximum plume), 3.1km (the local minimum), and 2.4km (the smaller local maximum). Back trajectories are run for 6 days for both isentropic (constant $\theta$) pathways and using the GDAS "model motion" (kinematic trajectories using vertical winds from the GDAS meteorology). For this case, at the two higher altitudes, these trajectories (while over the SEA Ocean) are remarkably similar to one another in terms of latitude and longitude, which allows us to explore the implications of each

configuration. For a given initial altitude, the two trajectories diverge in trajectory altitude, with the kinematic trajectories showing consistent subsidence (when looking forward in time) and the isentropic trajectories being fairly constant in altitude (at least after they depart the continent), but the two trajectories are very similar in terms of horizontal location, at least after exiting the continent (beyond that point, the trajectories become more uncertain due to convection over land).

Finally, the right-hand panel in Figure 16 shows these trajectories overlaid on the ERA5 reanalysis fields of water vapor (blue

shading) and potential temperatures ($\theta$, grey contours show isentropes at 3K intervals), following the location of the isentropic trajectories. Here we can clearly see the differences between the two trajectories are most pronounced in the vertical. We note that the isentropic trajectories as given by HYSPLIT (circle-lines) correspond to isentropic contours from ERA5 (grey curves) at all altitudes until the trajectory reaches (or rather, exits) the continent: on 8 September for the 4 km trajectory, and on 7 September for the 3.1 km trajectory. The 2.4 km trajectory is over the ocean during the entire trajectory and thus follows the

isentropes this entire period. Once trajectories are determined to be over the continent, they exhibit more variability in terms of altitude, as would be expected due to the strong convection in this region. This also likely indicates the trajectory analysis is less reliable beyond this point, but the trajectories are nonetheless consistent with airmasses originating from a diurnally-varying deep continental boundary layer.

The kinematic (nonisentropic) trajectories, in contrast, are seen to cross many $\theta$ curves during this time, but this is not

necessarily inconsistent with the ERA5 reanalysis: in terms of the water vapor, these back trajectories calculated using GDAS winds still remain within the humid layer for several days, until the trajectories are over the continent (4km and 3.1km on 8 and 7 September) or diverge from the isentropic trajectory (2.4km on 8 September). We note that the 2.4km trajectories diverged within 2 days of the analysis and the kinematic trajectory exits the top of this humid layer shortly thereafter; when the ERA5





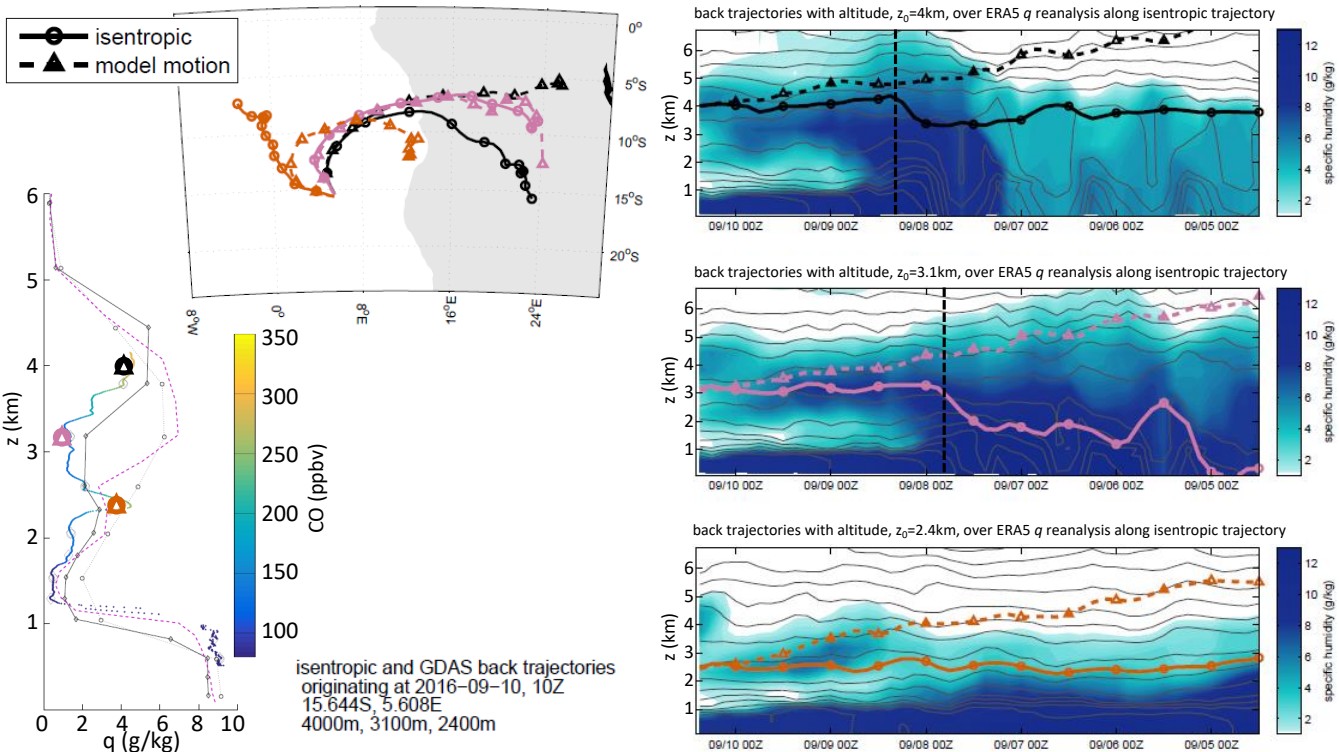

**Figure 16.** (bottom-left to right): An ORACLES aircraft profile from 10 September 2016 at approximately 10Z. Colors show the aircraft-measured CO corresponding to the measured water vapor, while the black solid, black dotted, and purple dashed lines show the ERA5, ERA-Interim, and MERRA-2 profiles at the same time and place. The map (top left-center) shows HYSPLIT back trajectories originating at three altitudes (bold shapes at bottom left) within this profile, and the right panel shows the ERA5 reanalysis $q$ profiles (blue shading) at the location and time of the isentropic paths, overlaid with the ERA5 potential temperatures ($\theta$, grey contours), and both the isentropic (solid/circle) and kinematic (dashed/triangle) HYSPLIT trajectories, for each altitude. Dashed vertical lines delineate when the 4km and 3.1km isentropic trajectories pass over the continent.

reanalysis is considered along the remainder of the 2.4km kinematic trajectory (i.e., at the HYSPLIT-indicated latitudes and longitudes), this trajectory too remains within the top of the water vapor plume until 3 September.

Taken together and considering the analytical caveats of each, these three perspectives on one sampling instance suggests that the airmass transport leading up to the aircraft observations may be somewhere in between the results of these two trajectories.

5 We remember that a too-strong subsidence is an issue in models over this region; Das et al. (2017) showed that vertical velocities in several different models were frequently too large compared with CALIOP satellite observations, especially once airmasses exit the continent. This is consistent with what we see here regarding very strong subsidence in the GDAS vertical motion, and suggests that the isentropic trajectories may be closer to the observed conditions. Yet the fact that the kinematic trajectories continue to follow the humid layer even with this strong subsidence indicates it is possible that these model trajectories are in

10 the famous model category of "wrong, but useful." Or rather, while the air masses sampled during ORACLES largely follow



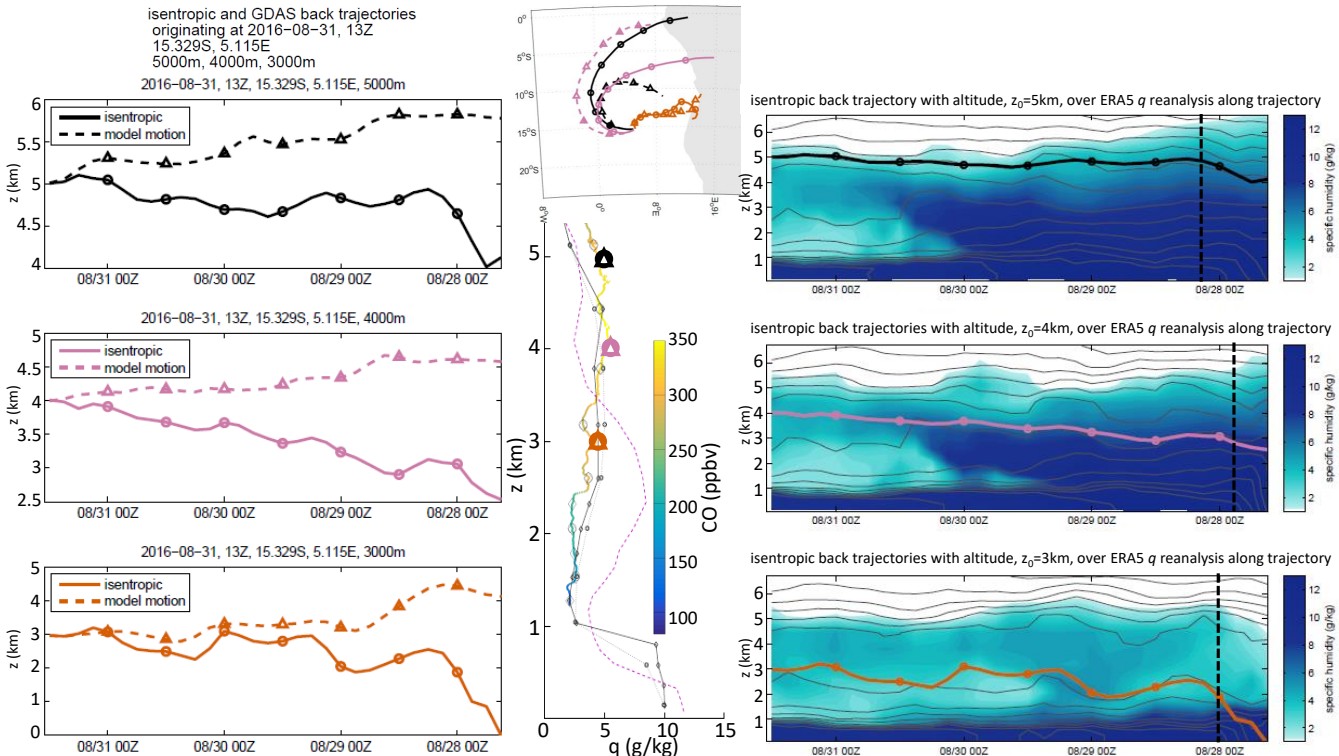

**Figure 17.** As in Figure 16, for three trajectories initialized from a profile from 31 August 2016 13Z which showed more uniform $q$ and CO with altitude than the profile on 10 September. Back trajectories are initialized at 3km, 4km, and 5km altitudes. As the isentropic and kinematic trajectories significantly diverge from one another in latitude and longitude, the trajectories to the right show ERA5 values along the isentropic trajectories only. The altitudes of both back trajectories in time are shown on the left.

isentropic pathways, there is some influence from clean, dry free tropospheric air especially over the continent. Indeed, this would be consistent with what we see in Figures 12 and 13: the linear relationship between CO and water vapor over the continent is largely driven by higher-altitude air which is only periodically influenced by continental sources; without these influences, the conditions of low-CO, low-$q$ would not be as prevalent in the air which is advected over the SEA.

5        As a final example, we consider the case shown in Figure 17, for back trajectories initialized at the aircraft profile sampled just before 13Z (12:35:21 to 12:50:14) on 31 August 2016, centered on 15.3°S, 5.1°E, in the same general area (Zone 3) as Figure 16. In contrast to the previous figure which was a very layered profile, this profile was fairly uniform in both $q$ and CO with altitude; this is corroborated by ERA5. Here, when we run the HYSPLIT back trajectories using model motion and isentropic motion initialized at three altitudes (3km, 4km, and 5km), we find that the two configurations diverge much more

10      rapidly. Again we find that the model motion trajectories (from GDAS meteorology) show very strong subsidence while the isentropic trajectories actually show the opposite: rising motion going forward in time. Spatially, the two trajectories diverge in latitude/longitude much earlier than did Figure 16, though both methods end up in largely the same location for the 4km and





5km trajectories. Looking at the ERA5 reanalysis along these trajectories, we find that the isentropic trajectories agree fairly well with the presence of the elevated water vapor plume, and some altitudes with fairly constant $\theta$, which may indicate these trajectories are less reliable, causing the discrepancy. This highlights the limitations of this type of analysis.

### 4.2 Sources of continental plume water vapor

We now briefly discuss the initial source of this continental water vapor. There are several potential explanations for the correlation between water vapor and the SEA BB plume, including direct emission of water vapor as a product of combustion; water vapor co-emission due to fuel properties; enhanced surface evaporation or evapotranspiration from the burning regions; or simple meteorological coincidence between plume air and already-humidified ambient air. As both smoke and water vapor have their source in the continental boundary layer, it may purely be coincidence of this source and further mixing with dry

and clean free-tropospheric air, but we briefly explore the other possibilities.

To the first point: some amount of water vapor is co-emitted with other gases and aerosols during combustion. Parmar et al. (2008) measured the ratio of enhanced water vapor to carbon dioxide and emissions $((\Delta H_2O)/(\Delta CO+\Delta CO_2))$ for different vegetation types: for savannah grasses this ratio is $\sim$1.2-1.6 and for some trees it reaches up to $\sim$3. For the sake of argument, even for a relatively high ratio of 3 (which should be an overestimate of the amount of water we should expect from burning

of savannah grass), this means that a 2 g/kg enhancement in water vapor would be accompanied by an enhancement of $\sim$ $800 - 1000$ppm of $\Delta CO+\Delta CO_2$.

For all three ORACLES deployments, the vast majority of $CO_2$ concentrations were measured as between 400 and 460 ppm and there were no measurements above 500 ppm. Based on these ratios and the $CO_2$ and water vapor concentrations observed during ORACLES, burning biomass could only have increased atmospheric water vapor by a tiny fraction of what

was observed. Unless either the estimates of the ratio of water vapor emitted per carbon dioxide and carbon monoxide or of the typical $\Delta CO+\Delta CO_2$ plume enhancement are too low by orders of magnitude, it is not plausible that the linear CO-$q$ relationship seen in ORACLES-2016 or the general moistness of the smoke plume are due to the co-emission of water vapor during biomass burning. The fact that the elevated water vapor ($\sim 2 - 4$g/kg) observed during ORACLES is not associated with a significant elevated $CO_2$ over the same region (on the order of 2000ppmv) suggests that the water vapor at least is not a

direct product of combustion.

Another possibility is that the moisture of the fuel itself could be evaporated during combustion; however, Potter (2005) suggested that for woody fuels, the fuel moisture would constitute no more than a third of the water vapor emitted by combustion, which would not account for the magnitude of the signal we observe. It is still plausible that some amount of the enhanced atmospheric water vapor near the fire sites could simply be a result of moist fuels releasing water vapor under the higher fire

temperatures; alternately, the observed $q$ could result entirely from surface evaporation/evapotranspiration independent of the fire conditions. Clements et al. (2006) also measured higher sensible and latent heat fluxes and increased turbulent mixing associated with the smoke plumes from small grass fires, and concluded that vapor emissions from such fires would have measurable impacts on local atmospheric dynamics, which may also be in play here. However, to these last points: since we find that models consistently reproduce some level of elevated $q$ without including either a source of water vapor co-emitted from



biomass burning, or an enhanced evaporation due to the higher surface temperatures in fire conditions, this suggests that these factors are not primary.

Thus, it seems likely that we can rule out direct co-emission of water vapor as the primary cause of the humid plume, and a meteorological coincidence seems to be the most likely explanation behind the observed correlations.

## 5   Conclusions

In the aircraft observations collected during the ORACLES field campaign over the southeast Atlantic Ocean, we find a robust correlation between plume strength, as indicated by both inlet-based CO concentration and column AOD, and water vapor concentration. The correlations are highly robust and linear in the September 2016 data and somewhat weaker in the more equatorial observations from August 2017 and October 2018. This could be due to a variety of factors, including the difference in season, deployment location, and sampling patterns over the SEA (e.g., routine diagonal versus routine north-south leg).

The ERA5 reanalysis is particularly accurate in placing its high humidity to be coincident with the higher humidity measured by ORACLES flights. All the other reanalyses/models showed a similar pattern, although these models show water vapor content which is more weakly correlated with $q$ from the aircraft observations. For the products which report CO, the CO-water vapor relationship shows the opposite pattern: the product which best corresponds to observed $q$ (WRF-Chem) shows the least consistent correlation between CO and $q$. In contrast, WRF-CAM5 and MERRA-2 both show somewhat better correlation between CO and $q$, but poorer correlation between modeled and observed $q$. This suggests that the CO-$q$ relationship overall is better represented in a free-running model (versus one which is frequently reinitialized) likely due to the different effect of this reinitialization on water vapor versus chemistry. However, such a free-running model results in a greater mismatch in the location of a given airmass compared with the observations (in latitude/longitude and in altitude).

On the regional scale, the ERA5 reanalysis shows humid air reaching high altitudes (700-500hPa; 3-6km) over the continent, albeit with a lag time from the surface. This is corroborated by other products. The analysis from MERRA-2 also indicates that the CO and $q$ are in phase with one another at the plume level, despite day to day variability in the actual magnitudes of each. Large-scale analysis thus suggests the air masses sampled over the ocean in ORACLES left the continent with the same relationship between water vapor and carbon monoxide as is observed by aircraft. This linear relationship develops over the continent due to diurnal upward mixing within the deep continental boundary layer (max height ~5-6km) which produces fairly consistent $q$ and CO vertical gradients (decreasing with altitude) which vary in time. Due to a combination of conditions including differential advection at different levels, daytime convection, nighttime subsidence, and resulting mixing between the smoky, moist continental boundary layer and the dry and fairly clean upper-troposphere air above ($\sim$ 6km), the vertically-aligned gradients effectively get stretched horizontally and into layer-like structures over the ocean. For conditions of strong zonal wind, the smoky, humid air is advected over the SEA following largely isentropic trajectories, where it persists, circulates, and in this case was sampled by ORACLES.

Water vapor, particularly when co-located with absorbing aerosols, will have significant impacts on both atmospheric radiative transfer (shortwave heating and longwave cooling) and cloud macrophysics and dynamics. An analysis which builds





upon our results here –and other components of the ORACLES dataset– to quantify the radiative impacts of this water vapor on the atmosphere over the broader SEA may thus help to clarify or corroborate previous studies of these effects. Future work will examine the year-to-year variation in this relationship, and the contributions of the BB plume and the humid layer to atmospheric radiative heating and aerosol-cloud interactions within this stratocumulus deck.

*Code and data availability.* The data used in this paper are publicly available at http://dx.doi.org/10.5067/Suborbital/ORACLES/P3/2016_V1 for the 2016 data and at http://dx.doi.org/10.5067/Suborbital/ORACLES/P3/2017_V1 and http://dx.doi.org/10.5067/Suborbital/ORACLES/P3/2018_V1 for the 2017 and 2018 data, respectively. The codes used in processing 4STAR data may be found at https://doi.org/10.5281/zenodo.1492912. ECMWF reanalyses are available at the Copernicus Climate Data Store (https://cds.climate.copernicus.eu/). HYSPLIT is available through the NOAA Air Resources Laboratory (https://www.ready.noaa.gov/HYSPLIT.php) and the compatible GDAS half-degree are found in the
gridded meteorological data archives (https://www.ready.noaa.gov/archives.php).

*Author contributions.* KP designed the research, performed the analysis, and wrote the paper with substantial input and feedback from PZ, RW, and MD. LP, J-MR, and RU contributed to the interpretation of the meteorological features and context. AMdS, PS, and GF contributed to the interpretation of the reanalysis and model results. AMdS provided the ORACLES-specific MERRA-2 reanalysis. PS provided the WRF-CAM5 model outputs. GF and GC provided the WRF-Chem model outputs. CF, SL, JR, MSR, and KP collected and processed the
4STAR data. JP collected and JP and YS processed the COMA data. DN collected and processed the WISPER data. ES collected and RB processed the onboard aircraft data. YS was ORACLES data manager and compiled the observations. JR, RW, and PZ were the ORACLES Principal Investigators. All authors were provided intermediate and final drafts of the manuscript for input and feedback.

*Competing interests.* The authors declare no competing interests.

*Acknowledgements.* ORACLES is funded by NASA Earth Venture Suborbital-2 grant NNH13ZDA001N-EVS2. The WISPER data was col-
lected with support from the NSF Atmospheric Chemistry and Climate and Large-scale Dynamics programs (grant number AGS 1564670). We thank the ORACLES deployment support teams, the ORACLES science team, and the governments and people of Walvis Bay and Swakopmund, Namibia, and São Tomé, São Tomé e Príncipe for a successful and productive mission.



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
