# Peer review of "Exploring the elevated water vapor signal associated with the free-tropospheric biomass burning plume over the southeast Atlantic Ocean"

_Atmospheric Chemistry and Physics, 2020_

## Author Comment (AC1)

*We thank both reviewers for their thoughtful replies. Our responses are indicated below in the italicized font, with the revised, tracked-changes draft attached. All line numbers below refer to this annotated revised draft.*

Reviewer #1

The manuscript titled "Exploring the elevated water vapor signal associated with the free-tropospheric biomass burning plume over the southeast Atlantic Ocean" by Kristina Pistone and co-authors investigated the association of CO-q with ORACLES aircraft data over SEA ocean. They have also analyzed the reanalysis and model simulations to understand the meteorological and dynamical dependence of BB plume-water vapour relationship. This manuscript is well written and scientifically sound. I recommend the publication of this manuscript in ACP.

> *We thank the reviewer for their comments.*

The explanation for the source of water vapour in the continental plume and its close association with BB CO and aerosols is not adequate. Since reanalysis and model results also showed similar variabilities, this could be of a meteorological coincidence rather than direct emissions. But why such a strong association exist between CO and q is not yet clear and needs to be explained in detail. Is there any study on source tagging or isotopic measurements of water vapour and aerosols over SEA?

> *Thank you for this comment. We had attempted to discuss in the paper that we do indeed believe that the correlation is due to essentially a meteorological coincidence (between diurnal heating, the timing of anthropogenic fires, and the diurnal cycle of the AEJ-S strength) and is not due to direct emissions from fires. We have edited the text to attempt to clarify this point (p. 34, Lines 6-10; p. 36, lines 10-11). While isotope measurements were made during ORACLES (e.g. Herman et al., 2020; https://doi.org/10.5194/amt-13-1825-2020, there has not yet been any source tagging analysis of these data.*

Authors discussed the influence of boundary layer evolution over land on the vertical transport of CO and water vapour over the continents. I could not find further quantitative supporting information on the boundary layer parameters (boundary layer height, fluxes: SHF and LHF etc) to supplement the arguments. Small write-up on the general boundary layer features and its diurnal structure during the BB events could be useful.

> *We have clarified this point in addressing the comments below regarding boundary layer height. We did not explicitly examine the modeled surface heat fluxes at this stage, as we felt the magnitudes and variations of the macrophysical parameters we considered (e.g., Fig 12) sufficiently illustrated the features we are describing here (i.e., diurnal heating). We have added additional text referring to another in-review work by Ryoo et al (2021), where they discuss the boundary layer height and other meteorological features in more detail. See p.24, Lines 31-34.*

The year-to-year variability of CO-q relationship is worth noting. Authors mentioned the airmass history over the BB regions, but more information is required on this point. Whether airmass pattern shows significant difference between 2016, 2017 and 2018? Notwithstanding the variability in time and meteorological conditions, what about the CO-q association for co-located measurements made during

2016, 2017 and 2018? Whether re-analysis and model simulations also depict weak association during 2017 and 2018?

*We agree that the differing airmass patterns for different years may be a significant factor regarding the different results. There are many potential contributing factors to why they're different, including year-to-year variability of the CO-q relationship, ranging from the large-scale variability to synoptic-, local-scale variability. Meteorology can be one factor. By dividing the 2017/2018 spatially in Figure 15 (colored vs grey), we attempted to address just one of these factors (spatial distribution) while explaining why in the present work we chose to focus only on exploring 2016. To fully explore the contributions of all the potential causes of these year-to-year differences is beyond the scope of this paper, but we hope to examine this more in future work. We have also added references to the newly-available paper from Ryoo et al (2021), which describes the differing meteorological contexts for each deployment month and how those compare to the monthly climatologies.*

Section 4.1 analyze the isentropic and kinematic airmass back trajectories using HYSPLIT. Though authors made broad comments on the usefulness and issues of back trajectory analysis, this section did not add more to the association of CO-q. Page 32, Line 1-4: This point is interesting, but needs more clarity and supporting evidence.

*We thank the reviewer for this comment. We can see how this section was indeed a bit disjointed from the rest of the analysis, although we do still believe it adds to the overall picture. In response, we've added a couple of sentences describing a similar analysis using MERRA-2 over ERA5; we did not include it in the actual paper as it's a bit difficult to assess the back trajectories along MERRA-2, due to the substantial vertical differences in the initial starting profile, but nonetheless it is consistent with our larger conclusions: for the model motion trajectories, which descend rapidly from high altitudes, the CO-q relationship along the MERRA-2 track is linear at each of these three points. We have added text to this effect on p. 34, Lines 6-10.*

There are several studies discussed the radiative implications of aerosols on water vapor and diurnal evolution of boundary layer over the continental locations. How does BB aerosols influence the CO-q association? I wonder whether diabatic heating due to absorbing aerosols has any effect on the elevated layers of water vapour. Moreover, photochemical oxidations and chemical reactions involving CH4 and OH can also affect the concentration of CO and water vapour. Though the strength of these mechanisms may not be adequate enough to explain the observed CO-q association, it is better to mention these possibilities in the discussion for the sake of completion.

*The reviewer brings up excellent points. We attempted to convey that diabatic heating due to aerosols at least appears to not be the \*cause\* of the elevated water vapor, as the ERA5 reanalysis would likely not capture these processes so accurately. While photochemical reactions are certainly important for aerosol conditions, including secondary organic formation and photochemical aging, we do not think this is relevant to CO on the timescales we consider (e.g. https://scied.ucar.edu/learning-zone/air-quality/carbon-monoxide). The heating \*resulting\* from the presence of elevated water vapor on aerosols/aerosols on water vapor (especially for an extended period of time during coincident transport from the continent over the ocean) may indeed have an impact on the ACI in the region. We did use this question as a partial motivation for this study, but in the interest of a more concise paper we have left an analysis of these impacts*

*for a future work. (Revised text: p. 2, Line 24 through p. 4, Line 3.)*

Figure 2 is interesting. Authors mentioned that the humidity datasets (aircraft, COMA, WISPER) differ for measurements within the PBL and rapidly changing aircraft conditions. I still not able to understand why only PBL humidity measured from the three instruments differ? What is the problem with PBL humidity and why aircraft is more stable (fewer ascents/descents) in the free troposphere compared to PBL. What is the rationale for omitting the PBL data is not clear? Is it possible to screen the data close to clouds?

> *We thank the reviewer for the opportunity to clarify this point. Our rationale for excluding PBL was twofold: first, as shown in Fig 2, the majority of the disagreement between instruments (grey points) occur both in the boundary layer (specifically at the boundary-layer top), during conditions of significant change in water vapor amount, or both. This is because the instruments were not completely directly co-incident (i.e., had different inlet lengths) and had different response times. During times of rapid descent into/ascent out of the boundary layer, the differences in instrumentation thus become very prominent in the 1Hz data. We have added more text to discuss the technical limitations of the instruments in certain cases, but these limitations do not have a significant effect on the subset of data on which we're currently focusing. (p. 10, Lines 10-17)*

> *To the second point, we omit the PBL data because in this paper we are explicitly concerned with the presence of water vapor within the BB plume itself. For lower altitudes (in or near to the PBL), the atmospheric water vapor is dominated by surface evaporation (i.e., ocean sources) which confuses the water vapor-CO relationship (It's not shown in Figure 3, but lower altitude water vapor exhibits no correlation with CO, but rather a range of q for a fairly constant background (~50ppbv) CO value). It is possible to screen closer to the top of the boundary layer (e.g., Ryoo et al. (2021), now cited in the paper) but this is not the focus of this present work.*

Page 11, Line 7-8: What is the rationale for selecting PBL height as 2 km? How do authors measure the PBL height (Page 12, Line 5)?

> *Establishing a definition of PBL height in ORACLES was actually quite a complex process: the published Redemann et al., (2021) contains cursory details, and Ryoo et al., (2021), currently in discussion, has a more in-depth definition and discussion of how PBL is calculated, and how that varies from the ERA5-defined PBL height. Both these papers are cited in the text. For simplicity of analysis, and because (as described above) we were interested primarily in the plume-level CO-q relationship, we chose a definite altitude cutoff rather than a dynamic definition of PBL as was used in these other analyses. Our main goal was to exclude the boundary-layer humid air from the continentally-sourced plume, and 2km accomplished this while still frequently including observations "below" the plume-proper. As seen in Figure 3, the range of CO and q is still quite large even with a fixed PBL cutoff, and Ryoo et al., (2021) shows the PBL height over ocean to be a maximum of 2km in the ORACLES region.*

> *We have edited this passage: "Note that while an actual determination of boundary layer height is more complex (as described by Ryoo et al., 2021), here we choose a simple cut of 2km as our goal is to focus on plume altitudes and exclude those with boundary layer influence."*

Page 13, Line 8-10: To assess the possibility of the hygroscopic growth of aerosols on the AOD versus q relationship, authors have to provide the ranges of relative humidity.

*Thank you for this comment. The majority of the inlet-based measurements were made at low RH, with half of the measurements at RH<40% (typically the "dry" threshold for in situ measurements) and an additional 30% between 40%<RH<60%. Less than 2% of the data considered were measured at RH>80%. We have added additional text to clarify this point (p. 14, Lines 14-23).*

Page 25, line 29: replace "continental boundary layer over land" with "continental boundary layer" (or boundary layer over land).

*Good point, we've reworded.*

Reviewer #2

In this study, the authors investigate the possible origin of the elevated moisture that characterises biomass-burning aerosol plumes transported from southern Africa to the south-eastern Atlantic Ocean. The authors find robust relationships between carbon monoxide (used as a marker of combustion sources) and specific humidity in ORACLES aircraft data and reanalyses and free-running models. They demonstrate that aircraft measurements are very probably real and that models simulate specific humidity sufficiently well for the analysis. They then use the models over land to track the source of the relationship, which they attribute to convection over the source regions of the biomass-burning aerosol. The moisture cannot originate from the fires themselves, which emit too small an amount of water to explain the measured enhancement.

The manuscript is very well written and follows a very clear reasoning. Figures support the discussion well. The paper makes a convincing case that elevated moisture and carbon monoxide are, to a large degree and despite their correlation, two independent quantities transported in the same air mass.

*We thank the reviewer for their time and their comments.*

My only main comment is to clarify the conclusion of the paper. In the conclusion section (Page 34 line 32 to page 35 line 4), and also – I think - at the end of the abstract (Page 2, lines 8-9), the authors call for more research on the effects of elevated water vapour on radiation and clouds. I am not sure why. I can see two direct implications of the results:

In that region, biomass-burning aerosols are transported in moist layers. That moisture modifies their aerosol optical depth and single-scattering albedo through hygroscopic growth, but such impacts are captured by aircraft and satellite retrievals.

The transported air layer would be moist even without fires. In other words, the elevated water vapour is part of the natural atmosphere and is not an external perturbation. Of course, it will have its own radiative effects and influence aerosol-cloud interactions by supplying moisture, but those impacts seem clearly identified, as discussed from Page 2 line 22 to Page 3 line 5.

In that context, what are the outstanding questions?

*Thank you for this comment. We concede that perhaps our framing was suboptimal, but as evidenced even by some of the comments of Reviewer 1, we think there is still much work to be done regarding this observed feature: for example, what drives the differences between the three deployments, and the impacts of heating a humid, aerosol-laden layer on the subsequent ACI. We*

*have edited Sections 1 and 3.5 to better convey this, and the last paragraph of Section 5 makes this more explicit.*

Other comments

Page 3, lines 28-31: These two statements suggest that aerosols need to be present for water vapour to influence clouds and have a radiative impact. Obviously, there is no need for aerosols for water vapour to interact with radiation. And lines 7-9 on the same page suggest that water vapour alone can exert radiative effects that are sufficiently strong to affect clouds. So I would suggest rewriting those statements.

*The reviewer brings up a fair point—we have revised this passage to clarify that certainly aerosols can exhibit these impacts alone; it now reads: "Taken together, these previous studies suggest that, first, the presence of above-cloud water vapor in conjunction with aerosol may modify the underlying cloud properties beyond solely the aerosol-induced semidirect effects, even without physically mixing into the cloud layer to alter the microphysics. Second, they suggest that the presence of water vapor associated with the presence of absorbing aerosol will impact radiative transfer of both longwave and shortwave through the atmospheric column."*

Page 8, line 13: I have never been sure of what aerosols do in the MERRA reanalysis. The MERRA papers say that GOCART is "radiatively coupled" to GEOS5 in that reanalysis, but does that then mean that aerosols affect heating rates?

*Yes, aerosol effects are fully included in the calculation of short- and longwave radiative fluxes in the model. MERRA-2 provides diagnostics of radiative fluxes with and without aerosol effects. This has been clarified in this passage: "MERRA-2 assimilates observations of meteorological parameters from multiple satellite platforms, as well as aerosol optical depth from satellites (MODIS, AVHRR) and ground-based (AERONET) measurements, into a comprehensive atmospheric model, with assimilated aerosol fields explicitly entering the calculation of radiative heating rates in the model."*

Page 12, line 7: "lower correlation than the other flights" – than the other routine flight? The flight on 20 September has an even lower correlation.

*Yes, the flight on the 20th was a flight of opportunity. We have edited this passage to clarify that the (routine) 25th has a lower correlation than the other routine flights, in addition to the two notably low correlations of the 20th and 24th flight of opportunities: "Regarding the routine flights, the flight on 25 September also has a lower correlation than the others."*

Page 12, line 18: I suggest clarifying that statement by echoing the statement made on page 20, lines 7-10, which is clearer on the assumption made: that because air masses are transported from the continent, one can reasonably extend the confidence gained over ocean to the continent.

*We're not sure which passage the reviewer is referring to here, as p. 12 doesn't have a line 18, but we have made minor edits throughout the paper to hopefully improve the clarity and readability of the final draft.*

Caption of Figure 12: "Each parameter has a distinct diurnal cycle except CO at 650hPa." What does that mean?

*This sentence has been clarified to: "The distinct diurnal cycle is seen for all variables except CO at 650hPa, where variability is dominated by multi-day changes rather than a strong diurnal cycle."*

Page 25, lines 1-4: Are those daily variations in CO emissions represented in MERRA2? If not, that might explain why the daily cycle of CO is flatter than other variables on Figure 12.

*The models, including MERRA-2, do account for the diurnal variation in emissions over this region. This has been clarified in Section 2.2.2: "MERRA-2 includes daily varying biomass burning emissions from Quick Fire Emission Dataset (QFED, Darmenov and da Silva 2015), with a prescribed diurnal cycle which peaks mid-afternoon."*

Page 27, line 4: What is meant by "background CO is used"? Doesn't the model simulate CO from emissions?

*Thank you for the comment. The chemical lifetime of CO in the atmosphere is around 3-4 months, so there is CO present and circulating globally which is not directly emitted from the fires under consideration; there is thus a background level of CO from longer-term circulation which must be taken into account in order to get accurate values. The sentence in question has been clarified to "…this is nonetheless consistent with the minimum CO observed by aircraft during ORACLES, suggesting the modeled background CO is accurate." (revision p. 27, Lines 19-20)*

Technical comments

Page 2, lines 31-32: Suggest rewriting to "the cloud liquid water path response to aerosol (via aerosol indirect effects) was much stronger".

*Thank you for the comment, this has been clarified.*

Page 3, line 7: The brackets in "(as was described by Adebiyi et al., 2015)" seem unnecessary.

*We have made this edit.*

Page 8, line 25, and Page 9, lines 4, 7, and 8: Is CAMS the Copernicus Atmosphere Monitoring Service Reanalysis (Inness et al. (2019) https://doi.org/10.5194/acp-19-3515-2019) or a typo for CAM5? Probably the former. Incidentally, the CAMS dataset could be an additional dataset in which to look at CO-q correlations. In fact, given its close link to ERA-5, CAMS may be a better dataset to do so than MERRA2.

*CAMS is the correct acronym in these instances. An analysis focusing on CAMS (which is of coarser resolution than ERA-5) is something that we didn't get to for this particular study but are interested to work on in future studies.*

Figure 9: The black dashed line that marks the African shoreline is not very visible. Could mark its position with an arrow on the x-axis?

*We have bolded the line in this figure.*

Page 23, line 23: Repeated word "the"

*Thank you, this has been corrected.*

[revised manuscript text omitted]